# A GEOMETRIC PERSPECTIVE ON VARIATIONAL AUTOENCODERS

## ABSTRACT

In this paper, we propose a geometrical interpretation of the Variational Autoencoder framework. We show that VAEs naturally unveil a Riemannian structure of the learned latent space. Moreover, we show that using these geometrical considerations can significantly improve the generation from the vanilla VAE which can now compete with more advanced VAE models on four benchmark datasets. In particular, we propose a new way to generate samples consisting in sampling from the uniform distribution deriving intrinsically from the Riemannian manifold learned by a VAE. We also stress the proposed method's robustness in the low data regime which is known as very challenging for deep generative models. Finally, we validate the method on a complex neuroimaging dataset combining both high dimensional data and low sample sizes.

## 1 INTRODUCTION

Variational Autoencoders (VAE) (Kingma & Welling, 2014; Rezende et al., 2014) are powerful generative models that map complex input data into a much lower dimensional space referred to as the latent space while driving the latent variables to follow a given prior distribution. Their simplicity to use in practice has made them very attractive models to perform various tasks such as high-fidelity image generation (Razavi et al., 2020), speech modeling (Blaauw & Bonada, 2016), clustering (Yang et al., 2019) or data augmentation (Chadebec et al., 2021).

Nonetheless, when taken in their simplest version, it was noted that these models produce most of the time blurry samples. This undesired behavior may be due to several limitations of the VAE framework. First, the training of a VAE aims at maximizing the Evidence Lower BOund (ELBO) which is only a lower bound on the true likelihood and so does not ensure that we are always actually improving the true objective (Burda et al., 2016; Alemi et al., 2016; Higgins et al., 2017; Cremer et al., 2018; Zhang et al., 2018). Second, the prior distribution used as regularization and for sampling may be too simplistic (Dai & Wipf, 2018) leading to poor data generation and there exists no guarantee that the actual distribution of the latent code will match a given prior distribution inducing over-regularization (Connor et al., 2021). Hence, trying to tackle those limitations through richer posterior distributions (Salimans et al., 2015; Rezende & Mohamed, 2015) or better priors (Tomczak & Welling, 2018) represents a major part of the proposed improvements over the past few years. However, the tractability of the ELBO constrains the choice in distributions and so finding a trade-off between model expressiveness and tractability remains crucial.

In this paper, we take a rather different approach and focus on the geometrical aspects a vanilla VAE is able to capture in its latent space. In particular, we propose the following contributions:

- We show that VAEs unveil naturally a latent space with a structure that can be modeled as a Riemannian manifold through the learned covariance matrices in the posterior distributions.

- We propose a natural sampling scheme consisting in sampling from a uniform distribution defined on the learned manifold and given by the metric. We show that this procedure improves significantly the generation process from a *vanilla* VAE and makes it able to perform as well as more advanced VAE models in terms of Frechet Inception Distance (Heusel et al., 2017) and Precision and Recall (Sajjadi et al., 2019) scores on four benchmark datasets.

- We also show that the propose method appears more robust to dataset size changes and outperforms even more strongly peers when only *smaller* sample sizes are considered.

- We validate the method on complex neuroimaging data from OASIS (Marcus et al., 2007).

## 2 VARIATIONAL AUTOENCODERS

Considering that we are given $x \in \mathcal{X}$ a set of data points deriving from an unknown distribution $p(x)$, a VAE aims at inferring such a distribution with a parametric model $\{p_\theta, \theta \in \Theta\}$ using a maximum likelihood estimator. A key assumption behind the VAE is to assume that the generation process also involves latent variables $\mathbf{z}$ living in a lower dimensional space such that the generative model writes

$$\mathbf{z} \sim q_{\mathrm{prior}}(\mathbf{z}) \quad ; \quad \mathbf{x} \sim p_\theta(\mathbf{x}|\mathbf{z}),$$

where $q_{\mathrm{prior}}$ is a prior distribution over the latent variables often taken as a standard Gaussian and $p_\theta(\mathbf{x}|\mathbf{z})$ is referred to as the decoder and is most of the time taken as a parametric distribution the parameters of which are estimated using neural networks. Hence, the likelihood $p_\theta$ writes:

$$p_\theta(x) = \int_{\mathcal{Z}} p_\theta(x|z)q(z)dz. \tag{1}$$

As this integral is most of the time intractable so is $p_\theta(\mathbf{z}|\mathbf{x})$, the posterior distribution. Hence, Variational Inference (Jordan et al., 1999) is used and a simple parametrized variational distribution $q_\phi(\mathbf{z}|\mathbf{x})$ is introduced to approximate the posterior $p_\theta(\mathbf{z}|\mathbf{x})$. $q_\phi(\mathbf{z}|\mathbf{x})$ is referred to as the *encoder* and, in the vanilla VAE, $q_\phi$ is chosen as a multivariate Gaussian whose parameters $\boldsymbol{\mu}_\phi$ and $\boldsymbol{\Sigma}_\phi$ are again given by neural networks. An unbiased estimate of the likelihood $p_\theta(\mathbf{x})$ can then be derived using importance sampling with $q_\phi(\mathbf{z}|\mathbf{x})$ and the ELBO objective follows using Jensen's inequality:

$$\log p_\theta(\mathbf{x}) = \log \mathbb{E}_{\mathbf{z} \sim q_\phi}\left[\hat{p}_\theta\right] \geq \mathbb{E}_{\mathbf{z} \sim q_\phi}\left[\log \hat{p}_\theta\right] = \underbrace{\mathbb{E}_{\mathbf{z} \sim q_\phi} \log p_\theta(\mathbf{x}|\mathbf{z}) - D_{\mathrm{KL}}(q_\phi(\mathbf{z}|\mathbf{x})\|p(\mathbf{z}))}_{ELBO} \tag{2}$$

The Evidence Lower BOund (ELBO) is now tractable since both $p_\theta(\mathbf{x}|\mathbf{z})$ and $q_\phi(\mathbf{z}|\mathbf{x})$ are parametrized and so can be optimized with respect to the *encoder* and *decoder* parameters.

**Remark 1** *In practice, $p_\theta(\mathbf{x}|\mathbf{z})$ is chosen depending on the modeling of the input data but is often taken as a simple distribution (e.g multivariate Gaussian, Bernoulli ...). Hence, the ELBO can also be seen as a two terms objective (Ghosh et al., 2020). The first one is a reconstruction term given by $p_\theta(\mathbf{x}|\mathbf{z})$ while the second one is a regularizer corresponding to the KL divergence between the posterior and the prior. For instance, in the case of a multivariate Gaussian we have*

$$\mathcal{L}_{REC} = \|\mathbf{x} - \boldsymbol{\mu}_\theta(\mathbf{z})\|_2^2, \quad \mathcal{L}_{REG} = D_{\mathrm{KL}}(q_\phi(\mathbf{z}|\mathbf{x})\|p(\mathbf{z})) \quad . \tag{3}$$

## 3 RELATED WORK

A natural way to improve the generation from VAEs consists in trying to use more complex priors (Hoffman & Johnson, 2016) than the standard Gaussian distribution used in the initial version such that they better match the true distribution of the latent codes. For instance, using a Mixture of Gaussian (Nalisnick et al., 2016; Dilokthanakul et al., 2017) or a Variational Mixture of Posterior (VAMP) (Tomczak & Welling, 2018) as priors was proposed. In the same vein, hierarchical latent variable models (Sønderby et al., 2016; Klushyn et al., 2019) or prior learning (Chen et al., 2016; Aneja et al., 2020) have recently emerged and aimed at finding the best suited prior distribution for a given dataset. Acceptance/rejection sampling method was also proposed to try to improve the expressiveness of the prior distribution (Bauer & Mnih, 2019). Some recent works linking energy-based models (EBM) and VAEs (Xiao et al., 2020) or modeling the prior as an EBM (Pang et al., 2020) have demonstrated promising results and are also worth citing .

On the ground that the latent space must adapt to the data as well, *geometry-aware* latent space modelings as hypershpere (Davidson et al., 2018), torus (Falorsi et al., 2018) or Poincaré disk (Mathieu et al., 2019) or discrete latent representations (Razavi et al., 2020) were proposed. Other recent contributions proposed to see the latent space as a Riemannian manifold where the Riemannian metric is given by the Jacobian of the generator function (Arvanitidis et al., 2018; Chen et al., 2018; Shao et al., 2018). This metric was then used directly within the prior modeled by Brownian motions

(Kalatzis et al., 2020). Others proposed to learn the metric directly from the data throughout training thanks to *geometry-aware* normalizing flows (Chadebec et al., 2020) or learn the latent structure of the data using transport operators (Connor et al., 2021). While these geometry-based methods show interesting properties of the learned latent space they either require the computation of a time consuming model-dependent function, the Jacobian, or add further parameters to the model to learn the metric or transport operators adding some computational burden to the method.

Arguing that VAEs are essentially autoencoders regularized with a Gaussian noise, Ghosh et al. (2020) proposed another interesting interpretation of the VAE framework and showed that other types of regularization may be of interest as well. Since the generation process from these autoencoders is no longer relying on the prior distribution, the authors proposed to use ex-post density estimation by fitting simple distributions such a Gaussian mixture in the latent space. While this paves the way for consideration of other ways to generate data, it mainly reduces the VAE framework to an autoencoder while we believe that it can also unveil interesting geometrical aspects.

Another widely discussed improvement of the model consists in trying to tweak the approximate posterior in the ELBO so that it better matches the true posterior using MCMC methods (Salimans et al., 2015) or normalizing flows (Rezende & Mohamed, 2015). For instance, methods using Hamiltonian equations in the flows to target the true posterior (Caterini et al., 2018) were proposed.

Finally, while discussing the potential link between PCA and autoencoders some intuitions arose on the impact of both the intrinsic structure of the variance of the data (Rakowski & Lippert, 2021) and the shape of covariance matrices in the posterior distributions (Rolinek et al., 2019) on disentanglement in the latent space. We also believe that these covariance matrices indeed play a crucial role in the modeling of the latent space but in this paper, we instead propose to see their inverse as the value of a Riemannian metric evaluated at the embedding points $\mu_i$.

## 4 PROPOSED METHOD

In this section, we argue that a vanilla VAE shows naturally a Riemannian structure of the latent space through the learned covariance matrices in the posterior distributions. We then propose a new natural generation scheme guided by this estimated geometry and consisting in sampling from a uniform distribution deriving intrinsically from the learned Riemannian manifold.

### 4.1 A WORD ON RIEMANNIAN GEOMETRY

First, we briefly recall some basic elements of Riemannian geometry needed in the rest of the paper. A more detailed discussion on integration and probability densities on manifolds may be found in Appendix A. A $d$-dimensional manifold $\mathcal{M}$ is a manifold which is locally homeomorphic to a $d$-dimensional Euclidean space. If the manifold $\mathcal{M}$ is further connected and differentiable it possesses a tangent space $T_{\boldsymbol{z}}$ at any $\boldsymbol{z} \in \mathcal{M}$ composed of the tangent vectors of the curves passing by $\boldsymbol{z}$. If $\mathcal{M}$ is equipped with a smooth inner product $g = \langle \cdot | \cdot \rangle_{\boldsymbol{z}}$ defined on its tangent space $T_{\boldsymbol{z}}$ for any $\boldsymbol{z} \in \mathcal{M}$ then $\mathcal{M}$ it is called a Riemannian manifold and $g$ is the associated Riemannian metric. Since $g$ is an inner product, a local representation of $g$ at any $\boldsymbol{z} \in \mathcal{M}$ is given by the positive definite matrix $\mathbf{G}(\boldsymbol{z})$. The notion of length of curves $\gamma : \mathbb{R} \to \mathcal{M}$ traveling in $\mathcal{M}$ can be defined as follows

$$L(\gamma) = \int_0^1 \sqrt{\langle \dot{\gamma}(t) | \dot{\gamma}(t) \rangle_{\gamma(t)}} dt = \int_0^1 \sqrt{\dot{\gamma}(t)^\top \mathbf{G}(\gamma(t)) \dot{\gamma}(t)} dt \,.$$

Curves minimizing $L$ are *geodesics* and a Riemannian distance between $\boldsymbol{z}_1, \boldsymbol{z}_2 \in \mathcal{M}$ can be defined

$$\text{dist}_{\mathbf{G}}(\boldsymbol{z}_1, \boldsymbol{z}_2) = \inf_\gamma L(\gamma) \quad \text{s.t.} \quad \boldsymbol{z}_1 = \gamma(0), \boldsymbol{z}_2 = \gamma(1) \,. \tag{4}$$

The manifold $\mathcal{M}$ is said to be *geodesically complete* if all geodesic curves can be extended to $\mathbb{R}$. In an Euclidean space, $\mathbf{G}$ reduces to the $\boldsymbol{I}_d$ and the distance becomes the classic Euclidean one.

**Remark 2** *A simple extension of this Euclidean framework consists in assuming that the metric is given by a constant positive definite matrix $\boldsymbol{\Sigma}$ different from $\boldsymbol{I}_d$. In such a case the induced Riemannian distance is the well-known Mahalanobis distance which writes*

$$\text{dist}_{\boldsymbol{\Sigma}} = \sqrt{(\boldsymbol{z}_2 - \boldsymbol{z}_1)^\top \boldsymbol{\Sigma} (\boldsymbol{z}_2 - \boldsymbol{z}_1)} \,.$$

## 4.2 THE RIEMANNIAN GAUSSIAN DISTRIBUTION

The notion of measure and so of probability distribution can be extended to *geodesically complete* Riemannian manifolds as well (Pennec, 2006). Given the Riemannian manifold $\mathcal{M}$ endowed with the Riemannian metric $\mathbf{G}$ and a chart $z$, an infinitesimal volume element may be defined on each tangent space $T_{\mathbf{z}}$ of the manifold $\mathcal{M}$ as follows

$$d\mathcal{M}_{\mathbf{z}} = \sqrt{\det \mathbf{G}(\mathbf{z})}dz\,, \tag{5}$$

with $dz$ being the Lebesgue measure. Hence, a Riemannian Gaussian distribution on $\mathcal{M}$ can be defined and consists in using the Riemannian distance defined in Eq. 4 instead of the Euclidean one

$$\mathcal{N}_{\mathrm{riem}}(\mathbf{z}|\sigma, \boldsymbol{\mu}) = \frac{1}{C}\exp\Big(-\frac{\mathrm{dist}_{\mathbf{G}}(\mathbf{z}, \boldsymbol{\mu})^2}{2\sigma}\Big), \quad C = \int_{\mathcal{M}}\exp\Big(-\frac{\mathrm{dist}_{\mathbf{G}}(\mathbf{z}, \boldsymbol{\mu})^2}{2\sigma}\Big)d\mathcal{M}_{\mathbf{z}}\,, \tag{6}$$

where $d\mathcal{M}_{\mathbf{z}}$ is the volume element defined by Eq. 5. Hence, the multivariate normal distribution is only a specific case of the Riemannian distribution with $\sigma = 1$, defined on the manifold $\mathcal{M} = \mathbb{R}^d$ endowed with the constant Riemannian metric $\mathbf{G}(\mathbf{z}) = \boldsymbol{\Sigma}^{-1}$, $\forall \mathbf{z} \in \mathcal{M}$.

## 4.3 GEOMETRICAL INTERPRETATION OF THE VAE FRAMEWORK

Within the VAE framework, the variational distribution $q_\phi(\mathbf{z}|\mathbf{x})$ is voluntarily chosen as a simple multivariate Gaussian distribution defined on $\mathbb{R}^d$ with $d$ being the latent space dimension. Hence, as explained in the previous section, given an input data point $\boldsymbol{x}_i$, the posterior $q_\phi(\mathbf{z}|\mathbf{x}) = \mathcal{N}(\boldsymbol{\mu}(\boldsymbol{x}_i), \boldsymbol{\Sigma}(\boldsymbol{x}_i))$ can also be seen as a Riemannian Gaussian distribution where the Riemannian distance is simply the distance with respect to the metric tensor $\boldsymbol{\Sigma}^{-1}(\boldsymbol{x}_i)$. Hence, the VAE framework can be seen as follows. As with an autoencoder, the VAE provides a code $\boldsymbol{\mu}(\boldsymbol{x}_i)$ which is a lower dimensional representation of an input data point $\boldsymbol{x}_i$. However, it also gives a tensor $\boldsymbol{\Sigma}^{-1}(\boldsymbol{x}_i)$ depending on $\boldsymbol{x}_i$ which can be seen as the value of a Riemannian metric $\mathbf{G}$ at $\boldsymbol{\mu}(\boldsymbol{x}_i)$ *i.e.*

$$\mathbf{G}(\boldsymbol{\mu}(\boldsymbol{x}_i)) = \boldsymbol{\Sigma}^{-1}(\boldsymbol{x}_i)\,.$$

This metric is crucial since it impacts the notion of distance in the latent space now seen as the Riemannian manifold $\mathcal{M} = (\mathbb{R}^d, \mathbf{G})$ and so changes the directions that are favored in the sampling from the posterior distribution $q_\phi(\mathbf{z}|\mathbf{x})$. Then, a sample $\mathbf{z}$ is drawn from a standard (*i.e.* $\sigma = 1$ in Eq. 6) Riemannian Gaussian distribution and fed to the decoder. As first approximation and since we only have access to a finite number of metric tensors $\boldsymbol{\Sigma}^{-1}(\boldsymbol{x}_i)$, the VAE model assumes that the metric is locally constant close to $\boldsymbol{\mu}(\boldsymbol{x}_i)$ and so the Riemannian distance reduces to the Mahalanobis distance in the posterior distribution. This simplified drastically the training process since now Riemannian distances have closed form and so are easily computable. Interestingly, the VAE framework will impose through the ELBO expression given in Eq. 3, that $\mathbf{z}$ gives a sample $\mathbf{x} \sim p_\theta(\mathbf{x}|\mathbf{z})$ close to $\boldsymbol{x}_i$ when decoded. Since $\mathbf{z}$ has a probability density function imposing higher probability for samples having the smallest Riemannian distance to $\boldsymbol{\mu}$, the VAE imposes in a way that latent variables that are close in the latent space with respect to the metric $\mathbf{G}$ will also provide samples that are close in the data space $\mathcal{X}$ in terms of L2 distance as noticed in Remark. 1. Noteworthy is that the latter distance can be amended through the choice of the decoder $p_\theta(\mathbf{x}|\mathbf{z})$. This is a an interesting property since it allows the VAE to directly link the learned Riemannian distance in the latent space to the distance in the data space. The regularization term in Eq. 3 ensures that the covariance matrices do not collapse to $\mathbf{0}_d$ and constraints the latent codes to remain close to the origin easing optimization. Finally, at the end of training, we have a lower dimensional representation of the training data given by the means of the posteriors $\boldsymbol{\mu}(\boldsymbol{x}_i)$ and a family of metric tensors $(\mathbf{G}_i = \boldsymbol{\Sigma}^{-1}(\boldsymbol{x}_i))$ corresponding to the value of a Riemannian metric defined locally on the latent space. Inspired from Hauberg et al. (2012), we propose to build a smooth continuous Riemannian metric defined on the entire latent space by performing the following interpolation:

$$\mathbf{G}(\mathbf{z}) = \sum_{i=1}^{N}\boldsymbol{\Sigma}^{-1}(\boldsymbol{x}_i) \cdot \omega_i(\mathbf{z}) + \lambda \cdot \boldsymbol{I}_d, \quad \omega_i(\mathbf{z}) = \exp\Big(-\frac{\mathrm{dist}_{\boldsymbol{\Sigma}^{-1}(\boldsymbol{x}_i)}(\mathbf{z}, \boldsymbol{\mu}_i)^2}{\rho^2}\Big), \tag{7}$$

where $\mathrm{dist}_{\boldsymbol{\Sigma}^{-1}(\boldsymbol{x}_i)}(\mathbf{z}, \boldsymbol{\mu}_i) = (\mathbf{z} - \boldsymbol{\mu}_i)^\top\boldsymbol{\Sigma}^{-1}(\boldsymbol{x}_i)(\mathbf{z} - \boldsymbol{\mu}_i)$ is the Riemannian distance between $\mathbf{z}$ and $\boldsymbol{\mu}_i$ with respect to the locally constant metric $\mathbf{G}(\boldsymbol{\mu}(\boldsymbol{x}_i)) = \boldsymbol{\Sigma}^{-1}(\boldsymbol{x}_i)$. Since the sum in Eq. 7

is made on the total number of training samples $N$, the number of reference metric tensors can be decreased for huge datasets by selecting only $k < N$ metric tensors[1] and increasing $\rho$ to reduce memory usage. We provide an ablation study on the impact of $\lambda$, the number of centroids $k$ and their choice along with a discussion on the choice for $\rho$ in Appendix G. Then, we have:

**Proposition 1** *The Riemannian manifold $\mathcal{M} = (\mathbb{R}^d, \mathbf{G})$ is geodesically complete.*

Prop. 1 (proved in Appendix B) allows now to refer to probability densities on $\mathcal{M}$. Rigorously, the metric defined in Eq. 7 should have been used during the training process. Nonetheless, this would have made the training longer and trickier since it would involve i) the computation of Riemannian distances that have no longer close form and so make the resolution of the optimization problem in Eq. 4 needed, ii) the sampling from Eq. 6 which is not trivial and iii) the computation of the regularization term. Instead, by approximating the value of the metric during training by its value at $\boldsymbol{\mu}_i$ (*i.e.* $\boldsymbol{\Sigma}_i^{-1}(x_i)$), the VAE training remains unchanged, stable and computationally reasonable since Riemannian Gaussians become multivariate Gaussians in $q_\phi(\mathbf{z}|\mathbf{x})$. Noteworthy is the fact that, likewise (Ghosh et al., 2020), in our vision of the VAE, the prior distribution is only seen as a regularizer though the KL term and other latent space regularization schemes may have been also envisioned. In the following, we keep the proposed vision and do not amend the training process.

### 4.4 GEOMETRY-AWARE SAMPLING

Assuming that the VAE has learned a latent representation of the data in a space seen as a Riemannian manifold, we propose to exploit this strong property to enhance the generation procedure. A natural way to sample from such a latent space would consist in sampling from the uniform distribution intrinsically defined on the learned manifold. Similar to the Gaussian distribution presented in Sec. 4.2, the notion of uniform distribution can indeed be extended to Riemannian manifolds. Given a bounded set $\mathcal{A} \subset \mathcal{M}$, the uniform distribution writes (Pennec, 2006)

$$p_{\mathcal{A}}(\mathbf{z}) = \frac{\mathbf{1}_{\mathcal{A}}(\mathbf{z})}{\text{Vol}(\mathcal{A})} = \frac{\mathbf{1}_{\mathcal{A}}(\mathbf{z})}{\int_{\mathcal{M}} \mathbf{1}_{\mathcal{A}}(z)d\mathcal{M}_z}\,.$$

This density is taken with respect to $d\mathcal{M}_z$, the Riemannian measure but using Eq. 5 and a coordinate system $z$ allows to obtain a pdf now defined with respect to the Lebesgue measure:

$$\mathcal{U}_{\text{Riem}}(\mathbf{z}) \propto \sqrt{\det \mathbf{G}(\mathbf{z})}\,.$$

Since the Riemannian metric has a closed form expression given by Eq. 7, sampling from this distribution is quite easy and may be performed using the HMC sampler (Neal, 2005) for instance. Now we are able to sample from the intrinsic uniform distribution which is a natural way of exploring the estimated manifold and the sampling is guided by the geometry of the latent space. A discussion on practical outcomes can be found in Appendix. C.

### 4.5 ILLUSTRATION ON A TOY DATASET

The usefulness of such sampling procedure may be easily appended in Figure 1 where a vanilla VAE was trained with a toy dataset composed of binary images of disks and rings of different size and thickness (example inspired by Chadebec et al. (2021)). On the left is presented the learned latent space along with the embedded training points given by the colored dots. The log of the metric volume element is given in gray scale. In this example, we clearly see a geometrical structure appearing since the disks and rings seem to wrap around each other. Obviously, sampling using the prior (taken as a $\mathcal{N}(0, \boldsymbol{I}_d)$) in such a case is far from being optimal since the sampling will be performed regardless of the underlying distribution of the latent variables and so will create irrelevant samples. To further illustrate this, we propose to interpolate between points in the latent space using different cost functions. Dashed lines represent affine interpolations while the solid ones show interpolation aiming at minimizing the potential $V(\boldsymbol{z}) = (\sqrt{\det \mathbf{G}(\boldsymbol{z})})^{-1}$ all along the curve *i.e.* solving the minimization problem

$$\inf_{\gamma} \int_0^1 V(\gamma(t))dt \quad \text{s.t.} \quad \gamma(0) = \boldsymbol{z}_1,\ \gamma(1) = \boldsymbol{z}_2\,. \tag{8}$$

---

[1]This may be performed with $k$-medoids algorithm for instance.

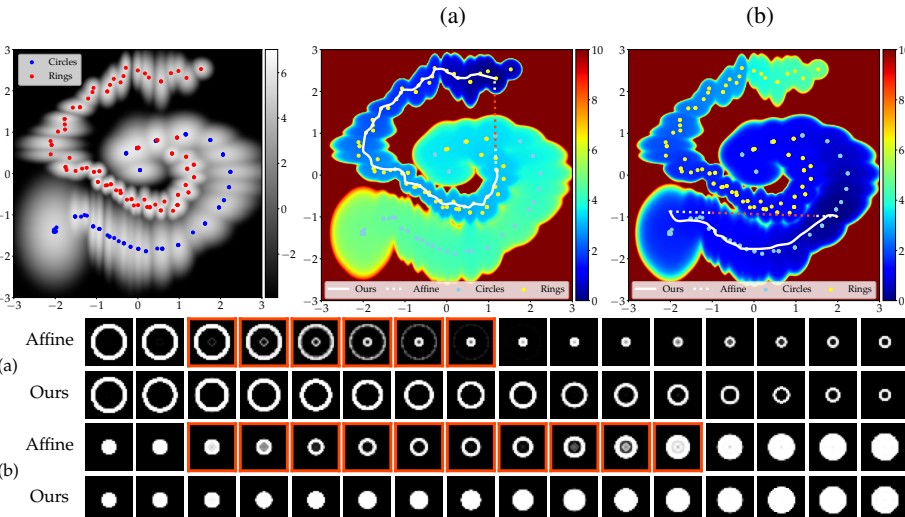

Figure 1: *Top:* Visualization and interpolation in a 2D latent space learned by a vanilla VAE trained with binary images of rings and disks. The log of the metric volume element $\sqrt{\det \mathbf{G}(z)}$ (also proportional to the log of the density we propose to sample from) is represented in gray scale. *Top right*: The Riemannian distance from a starting point is presented with the color maps. The dashed lines are affine interpolations between two points in the latent space and the solid ones are obtained by solving Eq. 8. *Bottom:* Decoded samples along the interpolation curves.

Below are presented the decoded samples all along the interpolation curves. Thanks to those interpolations we can see that i) the latent space seems to really have a specific geometrical structure since decoding all along the interpolation curves obtained by solving Eq. 8 leads to qualitatively satisfying results, ii) certain locations of the latent space must be avoided since sampling there will produce irrelevant samples (see red frames and corresponding red dashes). Using the proposed sampling scheme will allow to sample in the white areas and so ensure that the sampling remains close to the data *i.e.* where information is available and so does not produce irrelevant images when decoded.

## 5 EXPERIMENTS

In this section, we conduct a comparison with other VAE models using other regularization schemes, more complex priors, richer posteriors, ex-post density estimation or trying to take into account geometrical aspects including our method. In the following and to ensure a fair comparison, all the models share the same auto-encoding neural network architectures described in Appendix E.

### 5.1 GENERATION WITH BENCHMARK DATASETS

First, we compare the proposed sampling method to several VAE variants such as a Wasserstein Autoencoder (WAE) (Tolstikhin et al., 2018), Regularized Autoencoders (Ghosh et al., 2020) with either L2 decoder's parameters regularization (RAE-L2), gradient penalty (RAE-GP), spectral normalization (RAE-SN) or simple L2 latent code regularization (RAE), a vamp-prior VAE (VAMP) (Tomczak & Welling, 2018), a Hamiltonian VAE (HVAE) (Caterini et al., 2018), a geometry-aware VAE (RHVAE) (Chadebec et al., 2020) and an Autoencoder (AE). We elect these models since they use different ways to generate the data using either the prior or ex-post density estimation. For the latter, we use the approach of Ghosh et al. (2020) and fit a 10-component mixture of Gaussian in the latent space after training. The models are trained on MNIST (LeCun, 1998), SVHN (Netzer et al., 2011), CIFAR 10 (Krizhevsky et al., 2009) and CELEBA (Liu et al., 2015) and we keep the model achieving the best validation loss. See Appendix E for the comprehensive experimental setup. Figure 2 shows a qualitative comparison between the resulting generated samples for MNIST and CELEBA, the same plots are made available in Appendix D for SVHN and CIFAR 10. Interest-

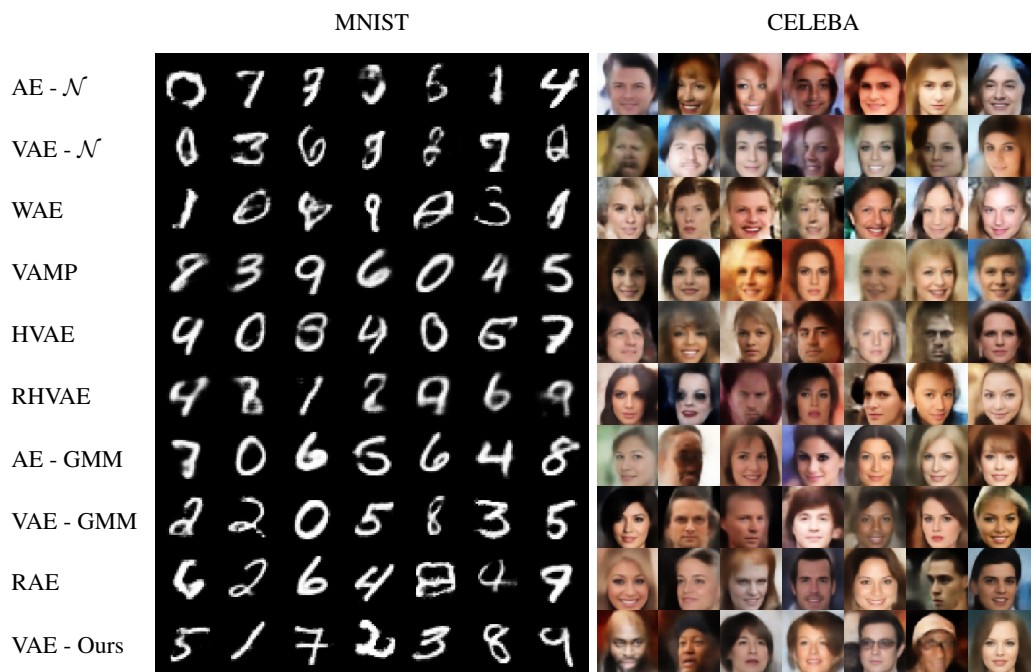

Figure 2: Generated samples with different models and generation processes. Generated samples with RAE variants are also provided in Appendix D.

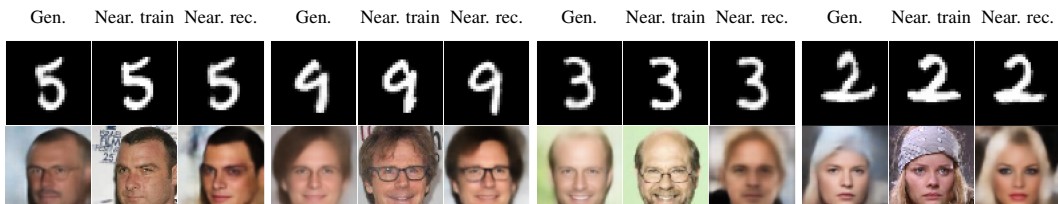

Figure 3: Nearest train image (near. train) and nearest image in all reconstructions of train images (near. recon) to the generated one (Gen.) with the proposed method. Note: the nearest reconstruction may be different from the reconstruction of the nearest train image.

ingly, using the non-prior based methods seems to produce qualitatively better samples (rows 7 to end). Nonetheless, the resulting samples seem even sharper when the sampling takes into account geometrical aspect of the latent space as we propose (last row). Additionally, even though the exact same model is used, we clearly see that using the proposed method represents a strong improvement of the generation process from a vanilla VAE when compared to the samples coming from a normal prior (second row). This insists on the fact that even the simplest VAE model actually contains a lot of information in its latent space but the limited expressiveness of the prior impedes to access to it. Hence, using more complex prior such as the VAMP may be a tempting idea. However, one must keep in mind that the ELBO objective in Eq. 2 must remain tractable and so using more expressive priors may be impossible. These observations are even more supported by Table 1 where we report the Frechet Inception Distance (FID) and the precision and recall (PRD) score against the test set to assess the sampling quality and diversity. Again, fitting a mixture of Gaussian (GMM) in the latent space appears to be an interesting idea since it allows for a better expressiveness and latent space prospecting. For instance, on MNIST the FID falls from 40.7 with the prior to 13.1 when using a GMM. Nonetheless, with the proposed method we are able to make it even smaller (8.5) and PRD scores higher without changing the model and performing post processing. This can also be observed on the 3 other datasets. Impressively, in almost all cases, the proposed generation method

Table 1: FID (lower is better) and PRD score (higher is better) for different models and datasets. In the first section are presented results using the prior distribution while in the second one, we use ex-post density estimation by fitting a 10-component mixture of Gaussian in the latent space.

| Model | MNIST (16) | | SVHN (16) | | CIFAR 10 (32) | | Celeba (64) | |
|---|---|---|---|---|---|---|---|---|
| | FID ↓ | PRD ↑ | FID ↓ | PRD ↑ | FID ↓ | PRD ↑ | FID ↓ | PRD ↑ |
| AE - $\mathcal{N}(0,1)$ | 46.41 | 0.86/0.77 | 119.65 | 0.54/0.37 | 196.50 | 0.05/0.17 | 64.64 | 0.29/0.42 |
| WAE | 20.71 | 0.93/0.88 | 49.07 | 0.80/**0.85** | 132.99 | 0.24/0.52 | 54.56 | **0.57**/0.55 |
| VAE - $\mathcal{N}(0,1)$ | 40.70 | 0.83/0.75 | 83.55 | 0.69/0.55 | 162.58 | 0.10/0.32 | 64.13 | 0.27/0.39 |
| VAMP | 34.02 | 0.83/0.88 | 91.98 | 0.55/0.63 | 198.14 | 0.05/0.11 | 73.87 | 0.09/0.10 |
| HVAE | 15.54 | 0.97/0.95 | 98.05 | 0.64/0.68 | 201.70 | 0.13/0.21 | 52.00 | 0.38/0.58 |
| RHVAE | 36.51 | 0.73/0.28 | 121.69 | 0.55/0.41 | 167.41 | 0.12/0.22 | 55.12 | 0.45/0.56 |
| AE - GMM | 9.60 | 0.95/0.90 | 54.21 | 0.82/0.83 | 130.28 | 0.35/0.58 | 56.07 | 0.32/0.48 |
| RAE (GP) | 9.44 | 0.97/**0.98** | 61.43 | 0.79/0.78 | 120.32 | 0.34/0.58 | 59.41 | 0.28/0.49 |
| RAE (L2) | 9.89 | 0.97/**0.98** | 58.32 | 0.82/0.79 | 123.25 | 0.33/0.54 | 54.45 | 0.35/0.55 |
| RAE (SN) | 11.22 | 0.97/**0.98** | 95.64 | 0.53/0.63 | 114.59 | 0.32/0.53 | 55.04 | 0.36/0.56 |
| RAE | 11.23 | **0.98**/0.98 | 66.20 | 0.76/0.80 | 118.25 | 0.35/0.57 | 53.29 | 0.36/0.58 |
| VAE - GMM | 13.13 | 0.95/0.92 | 52.32 | 0.82/**0.85** | 138.25 | 0.29/0.53 | 55.50 | 0.37/0.49 |
| VAE - Ours | **8.53** | **0.98**/0.97 | **46.99** | **0.84**/0.85 | **93.53** | **0.71**/0.68 | **48.71** | 0.44/**0.62** |

can either compete or outperform peers both in terms of FID and PRD scores. Finally, we check if the proposed method does not overfit the training data and is able to produce diverse samples by showing the nearest neighbor in the train set and the nearest image in all the reconstructions of the train images to a generated image in Figure 3. This experiment shows that the generated samples are not only resampled train images and that the sampling prospects quite well the manifold. To support even more this claim we provide in Appendix G an analysis in a case where only two centroids are selected in the metric. This also shows that the generated samples are not only an interpolation between the $k$ selected centroids since some generated images contain attributes that are not present in the images of the decoded centroids. The outcome of such an experiment is that using post training latent space processing such as ex-post density estimation or adding some geometrical consideration to the model allows to strongly improve the sampling without adding more complexity to the model. Generating 1000 samples on CELEBA takes approx. 5.5 min for our method vs. 4 min for a 10-component GMM and 0.6s for prior based methods on a single GPU V100-16GB.

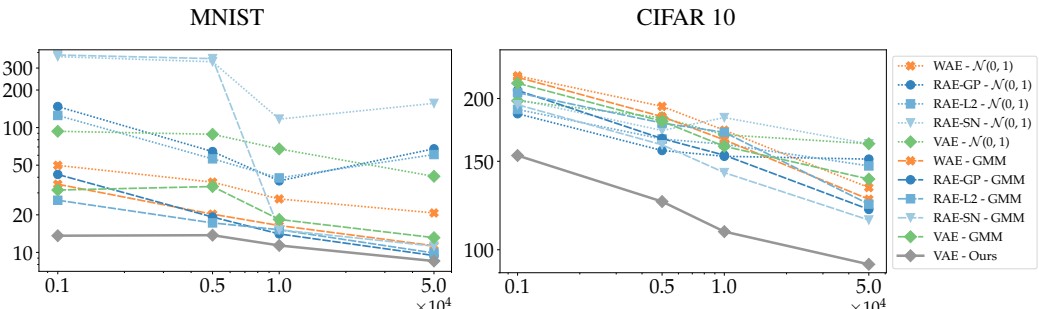

Figure 4: Evolution of the FID score according to the number of training samples.

## 5.2 INVESTIGATING GENERATION ROBUSTNESS IN LOW DATA REGIME

We perform a comparison using the same models and datasets as before but we decide to progressively decrease the size of the training set to see the robustness of the different sampling methods according to the number of samples. This experiment is rarely performed in most generative models related papers even though it is well known that such a context may be very challenging for these models. Nonetheless, it appears to us very important since in day-to-day applications collecting such large databases may reveal costly if not impossible (think of medicine for instance). Hence, we consider MNIST, CIFAR10 and SVHN and use either the full dataset size, 10k, 5k or 1k training samples. For each experiment, the best retained model is again the one achieving the best ELBO

Table 2: Classification results averaged on 20 independent runs. For the generative models, the classifier is trained on 2K generated samples per class.

| Generation method | Balanced Accuracy | F1 | |
|---|---|---|---|
| | | AD | CN |
| Original (unbalanced) | $66.2 \pm 7.6$ | $47.6 \pm 15.8$ | $87.3 \pm 2.0$ |
| Original (resampled) | $81.8 \pm 2.6$ | $72.1 \pm 3.6$ | $\mathbf{88.0 \pm 2.3}$ |
| AE - $\mathcal{N}(0, I_d)$ | $50.0 \pm 0.0$ | $0.0 \pm 0.0$ | $84.1 \pm 0.0$ |
| WAE | $57.4 \pm 9.7$ | $21.0 \pm 24.5$ | $84.4 \pm 2.3$ |
| VAE - $\mathcal{N}(0, I_d)$ | $51.8 \pm 3.8$ | $6.1 \pm 11.8$ | $84.6 \pm 1.1$ |
| VAMP | $83.1 \pm 2.6$ | $70.4 \pm 3.6$ | $82.2 \pm 4.7$ |
| HVAE | $56.3 \pm 7.9$ | $19.6 \pm 21.7$ | $85.4 \pm 1.7$ |
| RHVAE | $68.0 \pm 10.9$ | $47.0 \pm 24.2$ | $85.1 \pm 3.3$ |
| AE - GMM | $82.4 \pm 2.3$ | $69.5 \pm 3.1$ | $82.0 \pm 3.6$ |
| RAE (GP) | $63.9 \pm 9.8$ | $46.5 \pm 15.9$ | $70.6 \pm 19.6$ |
| RAE (L2) | $74.1 \pm 6.0$ | $60.6 \pm 9.5$ | $82.1 \pm 5.9$ |
| RAE (SN) | $62.3 \pm 8.9$ | $37.8 \pm 22.6$ | $80.1 \pm 7.9$ |
| RAE | $69.3 \pm 8.1$ | $53.8 \pm 12.9$ | $80.0 \pm 10.7$ |
| VAE - GMM | $83.0 \pm 3.6$ | $71.4 \pm 4.3$ | $85.3 \pm 3.0$ |
| VAE - Ours | $\mathbf{85.4 \pm 2.5}$ | $\mathbf{74.7 \pm 3.5}$ | $87.3 \pm 2.7$ |

on the validation set the size of which is set as 20% of the train size. See Appendix E for further details about experiments set-up. Then, we report the evolution of the FID against the test set in Figure 4. Results obtained on SVHN are presented in Appendix F. Again, the proposed sampling method appears quite robust to the dataset size since it outperforms the other models' FID even when the number of training samples is smaller. This is made possible thanks to the proposed metric that allows to avoid regions of the latent space having poor information. Finally, our study shows that although using more complex generation procedures such as ex-post density estimation seems to still enhance the generation capability of the model when the number of training samples remains quite high ($\geq$5k), this gain seems to worsen when the dataset size reduces as illustrated on CIFAR.

## 5.3 GENERATION WITH COMPLEX DATA

Finally, we also propose to stress the proposed generation procedure in a day-to-day scenario where the limited data regime is more than common. To stress the model in such condition, we consider the publicly available OASIS database composed of 416 MRI of patients, 100 of whom were diagnosed with Alzheimer disease (AD). Since both FID and PRD scores are not relevant and reliable in such low data regime due to the lack of a large test set, we propose to assess quantitatively the generation quality with a data augmentation task. Hence, we split the dataset into a train set (70%), a validation set (10%) and a test set (20%). Each model is trained on each label of the train set and used to generate 2k new samples per class. Then a simple CNN classifier is trained on i) the original train set and ii) the 4k generated samples from the generative models and tested on the test set. Table 2 shows the mean balanced accuracy and F1 scores across 20 runs. These metrics provide a good way to assess i) if the generative model can generate data that are not too far from the test set and add information to the data that is relevant for classification and ii) allows to quantify the amount of overfitting. The proposed method is the only one to be able to outperform the original (unbalanced) data both in terms of balanced accuracy and F1 scores for both labels meaning that generated samples are relevant to the classifier. This is also the sign of a good generalization power since the classifier achieves classification results that outperform the one observed on the original data.

## 6 CONCLUSION

In this paper, we provided a geometric understanding of the latent space learned by a VAE and showed that it can actually be seen as a Riemannian manifold. Then, we proposed a new natural generation process consisting in sampling from the intrinsic uniform distribution defined on this learned manifold. It showed to be competitive with more advanced versions of the VAEs using either more complex priors, ex-post density estimation, normalizing flows or other regularization schemes. Interestingly, the proposed method revealed good robustness properties in complex settings such as high dimensional data or low sample sizes. Future work would consist in trying to use this method to perform data augmentation in those challenging contexts and compare its reliability for such a task with state of the art augmentation methods.

## REPRODUCIBILITY STATEMENT

In order to make the method and the proposed experiments reproducible, we provide in Appendix E the complete experimental set-up and in Appendix C pseudo-code algorithms detailing the implementation from a practical point of view. We also provide an implementation in the supplementary material.

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

## A  FURTHER ELEMENTS ON RIEMANNIAN GEOMETRY

In the field of differential geometry, a Riemannian manifold $\mathcal{M}$ can be defined as a connected and differentiable manifold endowed with a Riemannian metric $g$. The metric $g$ is a smooth inner product $g : p \to \langle \cdot | \cdot \rangle_p$ defined on each tangent space $T_p\mathcal{M}$ of the manifold with $p \in \mathcal{M}$. A chart (or coordinate system) $(U, x)$ is a homeomorphism mapping an open set $U$ of the manifold to an open set $V$ of an Euclidean space. The manifold $\mathcal{M}$ is further called a $d-$dimensional manifold if for each chart $V \subset \mathbf{R}^d$. This means that there exists a neighborhood $U$ of each point $p \in \mathcal{M}$ such that $U$ is homeomorphic to $\mathbb{R}^d$. Given $p \in U$, a chart $\phi : (x^1, \ldots, x^d)$ induces a basis $\left( \frac{\partial}{\partial x^1}, \ldots, \frac{\partial}{\partial x^d} \right)_p$ on the tangent space $T_p\mathcal{M}$. Hence, the metric of a Riemannian manifold can be locally represented in the chart $\phi$ as a positive definite matrix $\mathbf{G}(p) = (g_{i,j})_{p,0 \leq i,j \leq d} = (\langle \frac{\partial}{\partial x^i} | \frac{\partial}{\partial x^j} \rangle_p)_{0 \leq i,j \leq d}$ for each point $p$ of the manifold. That is for $v, w \in T_p\mathcal{M}$ and $p \in \mathcal{M}$, the inner product writes $\langle u | w \rangle_p = u^\top \mathbf{G}(p) w$.

Two ways of apprehending manifolds exist. The first one is the extrinsic view and it assumes that the manifold is embedded within a higher dimensional Euclidean space. A simple example is the 2-dimensional sphere $\mathcal{S}^2$ seen as a subspace of $\mathbb{R}^3$. The second one which is adopted in this paper is the intrinsic view. In the intrinsic view the manifold is studied using its underlying structure and so the length of a curve $\gamma : \mathbb{R} \to \mathcal{M}$ traveling in the manifold cannot be interpreted using the Euclidean distance but requires to use the metric defined onto the manifold itself. Let $z_1, z_2 \in \mathcal{M}$ be two points of the manifold, and $\gamma$ be a curve traveling in $\mathcal{M}$ parametrized by $t \in [0, 1]$ such that $\gamma(0) = z_1$ and $\gamma(1) = z_2$. Then, the length of $\gamma$ is given by

$$\mathcal{L}(\gamma) = \int_0^1 \|\dot{\gamma}(t)\|_{\gamma(t)} dt = \int_0^1 \sqrt{\langle \dot{\gamma}(t) | \dot{\gamma}(t) \rangle_{\gamma(t)}} dt$$

Curves $\gamma$ that are minimizer(s) of this criteria are called *geodesic* curves. A distance dist on the manifold $\mathcal{M}$ can then be derived and writes:

$$\mathrm{dist}(z_1, z_2) = \min_\gamma \mathcal{L}(\gamma) \quad \text{s.t.} \quad \gamma(0) = z_1, \gamma(1) = z_2 \tag{9}$$

The manifold $\mathcal{M}$ is said to be *geodesically complete* if all geodesic curves can be extended to $\mathbb{R}$. For any $p \in \mathcal{M}$, the exponential map at $p$ maps a vector $v$ of the tangent space $T_p\mathcal{M}$ to a point of the manifold $\tilde{p} \in \mathcal{M}$ such that the geodesic starting at $p$ with initial velocity $v$ reached $\tilde{p}$ at time 1.

$$\mathrm{Exp}_p : \begin{cases} T_p\mathcal{M} & \to \mathcal{M} \\ v & \to \mathrm{Exp}_p(v) = \gamma_{(p,v)}(1) \end{cases},$$

where $\gamma_{(p,v)}(1)$ means $\gamma(0) = p$ and $\dot{\gamma}(0) = v$. Saying that the manifold $\mathcal{M}$ is *geodesically complete* means that the exponential maps is defined on the entire tangent space $T_p\mathcal{M}$ for each element $p \in \mathcal{M}$.

In Pennec (2006), the author discusses the statistical framework that can be developed on *geodesically complete* manifolds using an *intrinsic* point of view. In particular, given a positively oriented Riemannian manifold $\mathcal{M}$ and a chart $\phi = (x_1, \cdots, x_d)$, a volume form $\mathrm{dVol}_g$ can be defined as the $d$-form:

$$\mathrm{dVol}_g = \sqrt{\det(g_{i,j})}\, dx^1 \wedge \cdots \wedge dx^d$$

This represents an infinitesimal volume element on each tangent space and so a measure on the manifold $\mathcal{M}$

$$d\mathcal{M}(\phi^{-1}(x)) = \sqrt{\det g(\phi^{-1}(x))}\, dx$$

It follows that we are able to integrate functions $f : \mathcal{M} \supset U \to \mathbb{R}$ on a given chart $(U, \phi)$.

$$\int_U f(p) d\mathcal{M}(p) = \int_{\phi(U)} f(\phi^{-1}(x)) \sqrt{\det g(\phi^{-1}(x))}\, dx^1 \cdots dx^d$$

This notion can then be extended to the whole manifold $\mathcal{M}$ using partition of unity. In particular, such a property allows us to define probability distributions whose density is defined with respect to the measure on the manifold. We recall such definition from Pennec (2006) below

**Definition 1** *Let $\mathcal{B}(\mathcal{M})$ be the Borel $\sigma$-algebra of $\mathcal{M}$. The random point $\mathbf{p}$ has a probability density function $\rho_{\mathbf{p}}$ if:*

$$\forall \mathcal{X} \in \mathcal{B}(\mathcal{M}), \ \ \mathbb{P}(\mathbf{p} \in \mathcal{X}) = \int_{\mathcal{X}} \rho(p) d\mathcal{M}(p)$$

$$and \ \int_{\mathcal{M}} \rho(p) d\mathcal{M}(p) = 1$$

Finally, given a chart $\phi = (z_1, \cdots, z_n)$ defined on the whole manifold $\mathcal{M}$ and a random point $\mathbf{p}$ on $\mathcal{M}$, the point $\mathbf{z} = \phi(\mathbf{p})$ is a random point whose density $\rho'_{\mathbf{z}}$ may be written with respect to the Lebesgue measure as such (Pennec, 2006):

$$\rho'_{\mathbf{z}}(\mathbf{z}) = \rho_{\mathbf{p}}(\phi^{-1}(\mathbf{z}))\sqrt{\det g(\phi^{-1}(\mathbf{z}))} \tag{10}$$

# B   PROOF OF PROP. 1

We adapt the proof in Louis (2019) and Chadebec et al. (2021) to our specific metric. We will show that given the manifold $\mathcal{M} = \mathbb{R}^d$ and the Riemannian metric whose local representation is given by Eq. 7, any geodesic curve $\gamma :]a, b[\to \mathcal{M}$ is actually extensible to $\mathbb{R}$. Let us consider a geodesic curve $\gamma$ such that $\gamma$ cannot be extended to $\mathbb{R}$. There exists $I =]a, b[$ such that $I$ is the maximum definition domain of $\gamma$. We will show that with such an assumption we will end up with a contradiction. We recall the shape of the Riemannian metric:

$$\mathbf{G}(\boldsymbol{z}) = \sum_{i=1}^{N} \boldsymbol{\Sigma}^{-1}(\boldsymbol{x}_i) \cdot \omega_i(\boldsymbol{z}) + \lambda \cdot \boldsymbol{I}_d \,,$$

Since $\boldsymbol{\Sigma}_i$ are positive definite matrices we have $\boldsymbol{z}^\top \boldsymbol{\Sigma}_i \boldsymbol{z} > 0, \ \forall \, \boldsymbol{z} \in \mathcal{M} - \{0\}$. We further have $\omega_i(\boldsymbol{z}) > 0, \ \forall \, \boldsymbol{z} \in \mathcal{M}$ since the manifold is *geodesically complete*. Let $t_0 \in ]a, b[$ we therefore have for any $t \in ]a, b[$.

$$\lambda \cdot \|\dot{\gamma}(t)\|_2^2 \leq \lambda \cdot \|\dot{\gamma}(t)\|_2^2 + \sum_{i=1}^{N} \dot{\gamma}(t)^\top \boldsymbol{\Sigma}(\boldsymbol{x}_i)^{-1} \dot{\gamma}(t) \cdot \omega_i(\gamma(t))$$

$$\leq \|\dot{\gamma}(t)\|_{\gamma(t)}^2 = \|\dot{\gamma}(t_0)\|_{\gamma(t_0)}^2 \,,$$

where the last equality comes for the constant speed of geodesic curves (Carmo, 1992). Hence we have:

$$\|\gamma(t) - \gamma(t_0)\|_2 \leq \frac{\|\dot{\gamma}(t_0)\|_{\gamma(t_0)}}{\sqrt{\lambda}} \cdot |t - t_0| \,.$$

This means that for any $t \in ]a, b[$ the geodesic curve $\gamma$ remains within a compact set. We show now that the curve can actually be extended. Let us define the sequence $t_n \xrightarrow[n \to \infty]{} b$. Since the geodesic curves have a constant speed the set $I = \{(t_n, \dot{\gamma}(t_n)\}_{n \in \mathbb{N}}$ is compact. Moreover, using Cauchy-Lipchitz theorem, we can find $\varepsilon > 0$ such that for any $n \in \mathbb{N}$, the geodesic $\gamma$ can be extended to $]t_n - \varepsilon, t_n + \varepsilon[$. Since, $t_n$ can be as close as $b$ as desired we can assure that the curve definition domain can be extended to $]a, b + \frac{\varepsilon}{2}[$.

## C  THE GENERATION PROCESS ALGORITHM - IMPLEMENTATION DETAILS

In this appendix, we provide pseudo-code algorithms explaining how to build the metric from a trained VAE and how to use the proposed sampling process. Noteworthy is the fact that we do not amend the training process of the vanilla VAE which remains pretty simple and stable.

### C.1  BUILDING THE METRIC

In this section, we explain how to build the proposed Riemannian metric. For the sake of clarity, we recall the expression of the metric in Eq. 7 below

$$\mathbf{G}(\boldsymbol{z}) = \sum_{i=1}^{N} \boldsymbol{\Sigma}^{-1}(\boldsymbol{x}_i) \cdot \omega_i(\boldsymbol{z}) + \lambda \cdot \boldsymbol{I}_d \,,$$

where

$$\omega_i(\boldsymbol{z}) = \exp\left( - \frac{\mathrm{dist}_{\boldsymbol{\Sigma}^{-1}(\boldsymbol{x}_i)}(\boldsymbol{z}, \boldsymbol{\mu}_i)^2}{\rho^2} \right) = \exp\left( - \frac{(\boldsymbol{z} - \boldsymbol{\mu}_i)^\top \boldsymbol{\Sigma}^{-1}(\boldsymbol{x}_i)(\boldsymbol{z} - \boldsymbol{\mu}_i)}{\rho^2} \right),$$

---

**Algorithm 1** Building the Metric from a Trained Model
---

**Input:** A trained VAE model $m$, the training dataset $\mathcal{X}$, $\lambda$
**for** $\boldsymbol{x}_i \in \mathcal{X}$ **do**
    $\boldsymbol{\mu}_i, \boldsymbol{\Sigma}_i = m(\boldsymbol{x}_i)$                 ▷ Retrieve training embeddings and covariance matrices
**end for**
Select $k$ centroids $\boldsymbol{c}_i$ in the $\boldsymbol{\mu}_i$                 ▷ e.g. with $k$-medoids
Get corresponding covariance matrices $\boldsymbol{\Sigma}_i$
$\rho \leftarrow \max_i \min_{j \neq i} \|\boldsymbol{c}_i - \boldsymbol{c}_j\|_2$       ▷ Set $\rho$ to the max distance between two closest neighbors
Build the metric using Eq. 7

$$\mathbf{G}(\boldsymbol{z}) = \sum_{i=1}^{N} \boldsymbol{\Sigma}_i^{-1} \cdot \omega_i(\boldsymbol{z}) + \lambda \cdot \boldsymbol{I}_d$$

**Return G**                                           ▷ Return **G** as a function

---

As is standard in VAE implementations, we assume that the covariance matrices $\boldsymbol{\Sigma}_i$ given by the VAE are diagonal and that the encoder outputs a mean vector and the log of the diagonal coefficients. In the implementation, the exponential is then applied to recover the $\boldsymbol{\Sigma}_i$ so that no singular matrix arises.

### C.2  SAMPLING PROCESS

Further to the description performed in the paper, we provide here a detailed algorithm stating the main steps of the generation process.

#### C.2.1  THE HMC SAMPLER

In the sampling process we propose to rely on the Hamiltonian Monte Carlo sampler to sample from the Riemanian uniform distribution. In a nutshell, the HMC sampler aims at sampling from a target distribution $p_{\mathrm{target}}(\boldsymbol{z})$ with $\boldsymbol{z} \in \mathbb{R}^d$ using Hamiltonian dynamics. The main idea behind such a sampler is to introduce an auxiliary random variable $\boldsymbol{v} \sim \mathcal{N}(0, I_d)$ independent from $\boldsymbol{z}$ and mimic the behavior of a particle having $\boldsymbol{z}$ (resp. $\boldsymbol{v}$) as location (resp. velocity). The Hamiltonian of the particle then writes

$$H(\boldsymbol{z}, \boldsymbol{v}) = U(\boldsymbol{z}) + K(\boldsymbol{v}) \,,$$

where $U(\boldsymbol{z})$ is the potential energy of such a particle and $K(\boldsymbol{v})$ is its kinetic energy both given by

$$U(\boldsymbol{z}) = -\log p_{\mathrm{target}}(\boldsymbol{z}), \qquad K(\boldsymbol{v}) = \frac{1}{2}\boldsymbol{v}^\top \boldsymbol{v}$$

The following Hamilton's equations govern the evolution in time of the particle.

$$\begin{cases} \frac{\partial H(\boldsymbol{z},\boldsymbol{v})}{\partial \boldsymbol{v}} &= \boldsymbol{v}\,, \\ \frac{\partial H(\boldsymbol{z},\boldsymbol{v})}{\partial \boldsymbol{z}} &= -\nabla_{\boldsymbol{z}} \log p_{\text{target}}(\boldsymbol{z})\,. \end{cases} \tag{11}$$

In order to integrate these equations, recourse to the leapfrog integrator is needed and consists in applying $n_{\text{lf}}$ times the following equations.

$$\begin{cases} \boldsymbol{v}(t + \frac{\varepsilon_{\text{lf}}}{2}) &= \boldsymbol{v}(t) + \frac{\varepsilon_{\text{lf}}}{2} \cdot \nabla_{\boldsymbol{z}} \log p_{\text{target}}(\boldsymbol{z}(t))\,, \\ \boldsymbol{z}(t + \varepsilon_{\text{lf}}) &= \boldsymbol{z}(t) + \varepsilon_{\text{lf}} \cdot \boldsymbol{v}(t + \frac{\varepsilon_{\text{lf}}}{2})\,, \\ \boldsymbol{v}(t + \varepsilon_{\text{lf}}) &= \boldsymbol{v}(t + \frac{\varepsilon_{\text{lf}}}{2}) + \frac{\varepsilon_{\text{lf}}}{2} \cdot \nabla_{\boldsymbol{z}} \log p_{\text{target}}(\boldsymbol{z}(t + \varepsilon_{\text{lf}}))\,, \end{cases} \tag{12}$$

where $\varepsilon_{\text{lf}}$ is called the leapfrog step size. This algorithm produces a proposal $(\widetilde{\boldsymbol{z}}, \widetilde{\boldsymbol{v}})$ that is accepted with probability $\alpha$ where

$$\alpha = \min\left(1, \exp\left(H(\boldsymbol{z},\boldsymbol{v}) - H(\widetilde{\boldsymbol{z}}, \widetilde{\boldsymbol{v}})\right)\right)\,.$$

This procedure is then repeated to create an ergodic Markov chain $(\boldsymbol{z}^n)$ converging to the distribution $p_{\text{target}}$ (Neal & others, 2011; Duane et al., 1987; Liu, 2008; Girolami & Calderhead, 2011).

## C.3 THE PROPOSED ALGORITHM

In our setting the target density is given by the density of the Riemannian uniform distribution which writes with respect to Lebesgue measure as follows

$$p(\mathbf{z}) = \mathcal{U}_{\text{Riem}}(\mathbf{z}) = \frac{1}{C} \sqrt{\det \mathbf{G}(\mathbf{z})}\,. \tag{13}$$

The log density follows

$$\log p(\mathbf{z}) = \frac{1}{2} \log \det \mathbf{G}(\mathbf{z}) - \log C\,,$$

In such a case, the Hamiltonian writes

$$H(\boldsymbol{z},\boldsymbol{v}) = -\log p(\boldsymbol{z}) + \frac{1}{2}\boldsymbol{v}^{\top}\boldsymbol{v}\,,$$

and Hamilton's equations become

$$\begin{cases} \frac{\partial H(\boldsymbol{z},\boldsymbol{v})}{\partial \boldsymbol{v}} &= \boldsymbol{v}\,, \\ \frac{\partial H(\boldsymbol{z},\boldsymbol{v})}{\partial \boldsymbol{z}_i} &= -\frac{\partial \log p(\boldsymbol{z})}{\partial \boldsymbol{z}_i} = -\frac{1}{2}\text{tr}\left(\mathbf{G}^{-1}(\boldsymbol{z})\frac{\partial \mathbf{G}(\boldsymbol{z})}{\partial \boldsymbol{z}_i}\right) \end{cases}$$

Since the covariance matrices are supposed to be diagonal as is standard in VAE implementations, the computation of the inverse metric is straightforward. Moreover, since $\mathbf{G}(\boldsymbol{z})$ is smooth and has a closed form, it can be differentiated with respect to $\boldsymbol{z}$ pretty easily. Now, the leapfrog integrator given in Eq. 12 can be used and the acceptance ratio $\alpha$ is easy to compute. Noteworthy is the fact that the normalizing constant $C$ is never needed since it vanishes in the gradient computation and simplifies in the acceptance ratio $\alpha$. We provide a pseudo-code of the proposed sampling procedure in Alg. 2. A typical choice in the sampler's hyper-parameters used in the paper is $N = 100$, $n_{\text{lf}} = 10$ and $\varepsilon_{\text{lf}} = 0.01$. The initialization of the chain can be done either randomly or on points that belong to the manifold (i.e. the centroids $\boldsymbol{c}_i$ or $\mu_i$).

---

**Algorithm 2** Proposed Sampling Process

---

**Input:** The metric function $\mathbf{G}$, hyper-parameters of the HMC sampler (chain length $N$, number of leapfrog steps $n_{\mathrm{lf}}$, leapfrog step size $\varepsilon_{\mathrm{lf}}$)

**Initialization:** $\boldsymbol{z}$             $\triangleright$ Initialize the chain

**for** $i = 1 \rightarrow N$ **do**

     $\boldsymbol{v} \sim \mathcal{N}(0, I_d)$           $\triangleright$ Draw a velocity

     $H_0 \leftarrow H(\boldsymbol{z}, \boldsymbol{v})$           $\triangleright$ Compute the starting Hamiltonian

     $\boldsymbol{z}_0 \leftarrow \boldsymbol{z}$

     **for** $k = 1 \leftarrow n_{\mathrm{lf}}$ **do**

         $\bar{\boldsymbol{v}} \leftarrow \boldsymbol{v} - \frac{\varepsilon_{\mathrm{lf}}}{2} \cdot \nabla_{\boldsymbol{z}} H(\boldsymbol{z}, \boldsymbol{v})$

         $\widetilde{\boldsymbol{z}} \leftarrow \boldsymbol{z} + \varepsilon_{\mathrm{lf}} \cdot \bar{\boldsymbol{v}}$           $\triangleright$ Leapfrog step Eq. 12

         $\widetilde{\boldsymbol{v}} \leftarrow \bar{\boldsymbol{v}} - \frac{\varepsilon_{\mathrm{lf}}}{2} \cdot \nabla_{\boldsymbol{z}} H(\widetilde{\boldsymbol{z}}, \bar{\boldsymbol{v}})$

         $\boldsymbol{v} \leftarrow \widetilde{\boldsymbol{v}}$

         $\boldsymbol{z} \leftarrow \widetilde{\boldsymbol{z}}$

     **end for**

     $H \leftarrow H(\widetilde{\boldsymbol{z}}, \widetilde{\boldsymbol{v}})$           $\triangleright$ Compute the ending Hamiltonian

     Accept $\widetilde{\boldsymbol{z}}$ with probability $\alpha = \min\left(1, \exp(H_0 - H)\right)$

     **if** Accepted **then**

         $\boldsymbol{z} \leftarrow \widetilde{\boldsymbol{z}}$

     **else**

         $\boldsymbol{z} \leftarrow \boldsymbol{z}_0$

     **end if**

**end for**

**Return** $\boldsymbol{z}$

---

# D    OTHER GENERATION

## D.1    SOME FURTHER SAMPLES ON CELEBA AND MNIST

In this section, we provide some further generated samples using the proposed method. Figure 5 and Figure 6 again support the fact that the method is able to generate sharp and diverse samples. We also add the other variants of the RAE model in Figure 7.

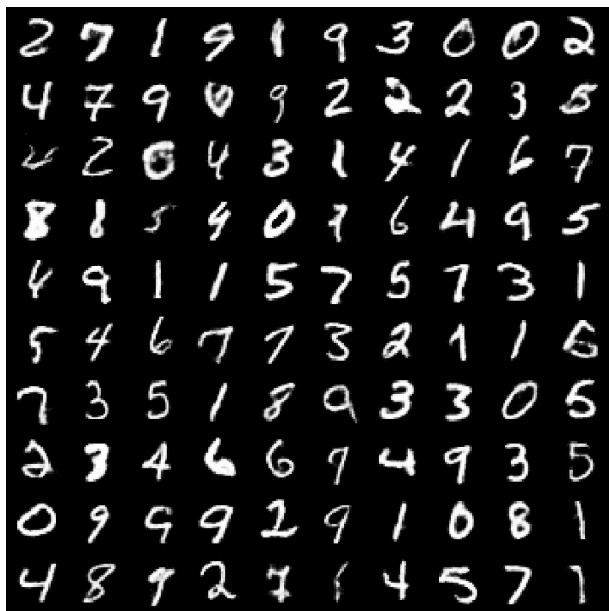

Figure 5: 100 samples with the proposed method on MNIST dataset.

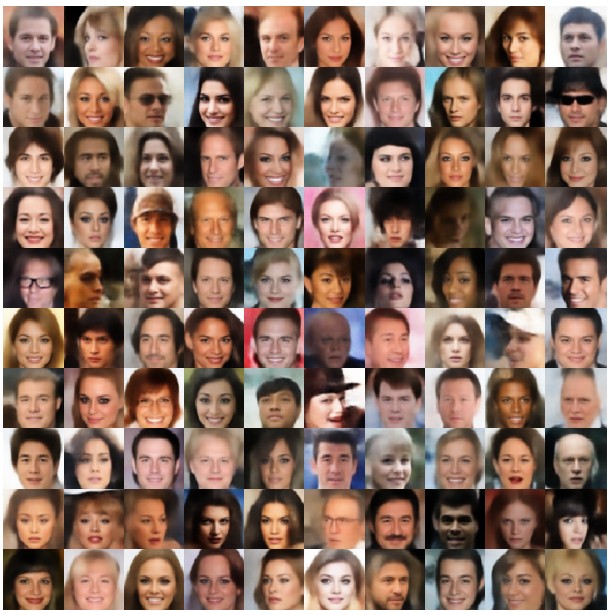

Figure 6: 100 samples with the proposed method on Celeba dataset.

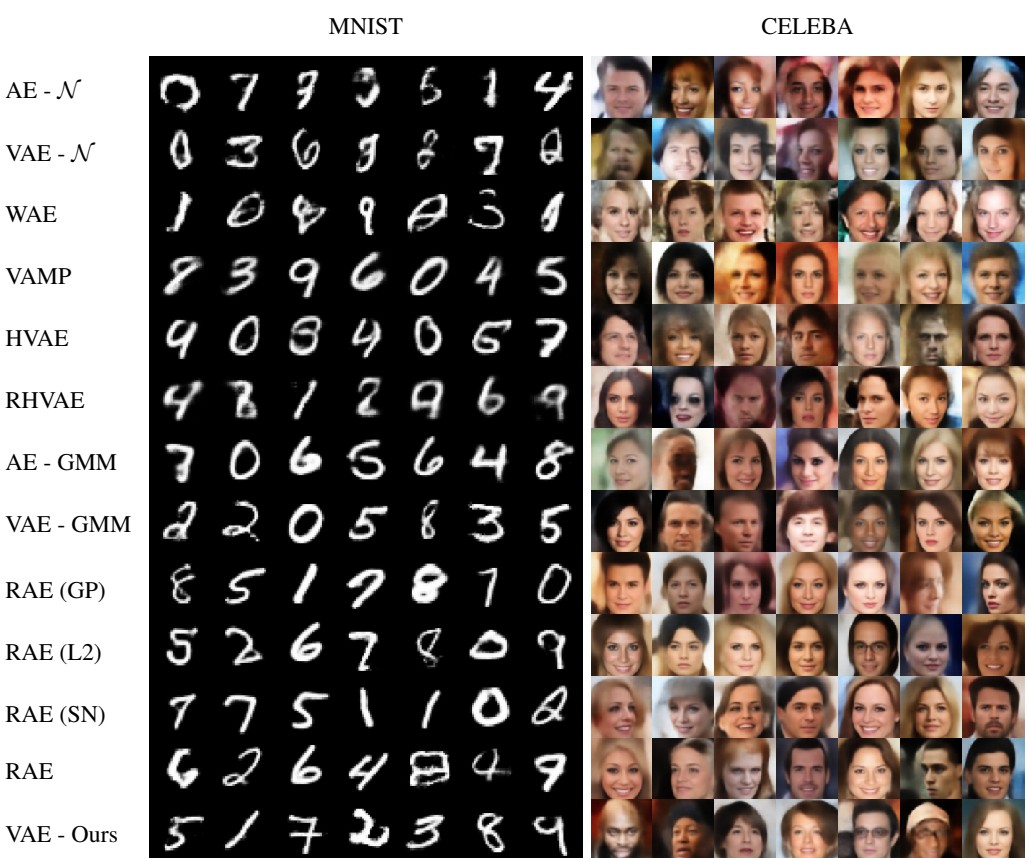

Figure 7: Generated samples with different models and generation processes.

## D.2 CIFAR AND SVHN

In this appendix, we gather the resulting samplings from the different considered models for SVHN and CIFAR 10.

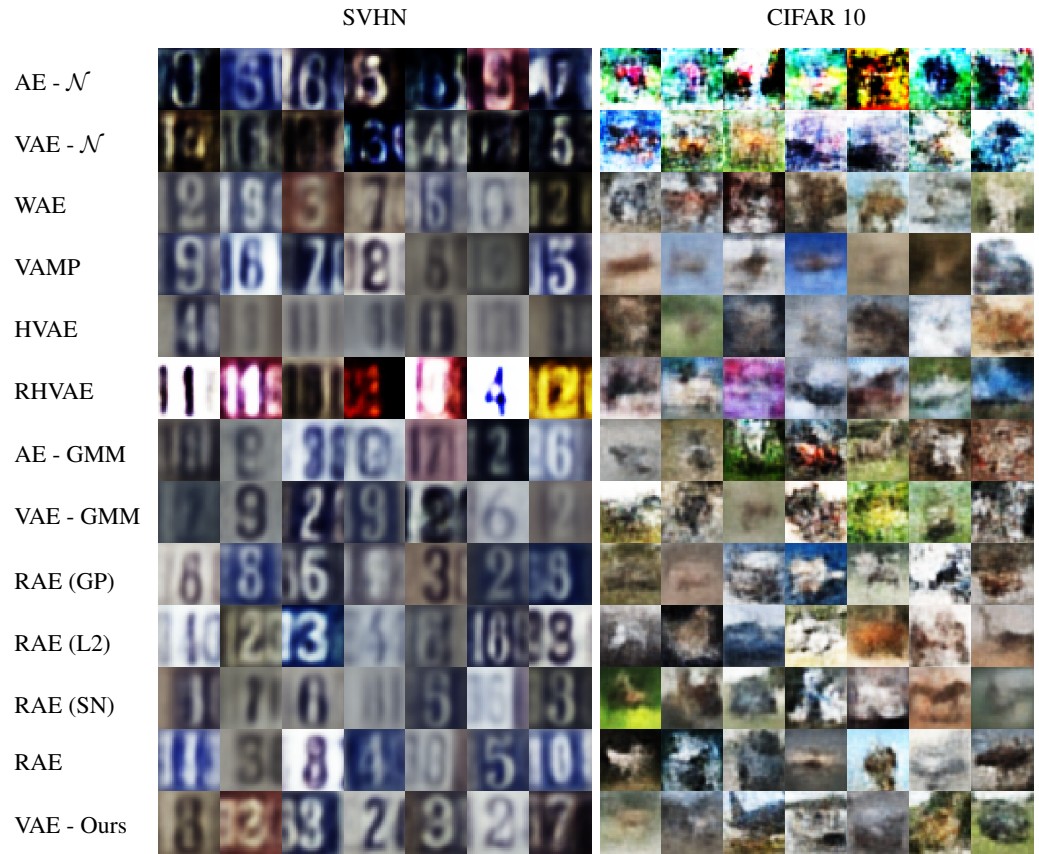

Figure 8: Generated samples with different models and generation processes.

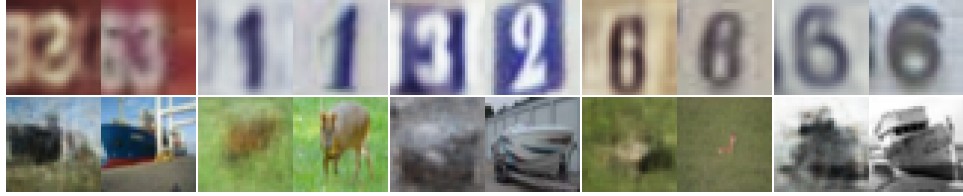

Figure 9: Closest element in the training set (Near.) to the generated one (Gen.) with the proposed method.

# E    EXPERIMENTAL SET-UP

The RAEs, VAEs and AEs are trained for 100 epochs for SVHN, MNIST[2] and Celeba and 200 on CIFAR10. Each time we use the official train and test split of the data. For MNIST and SVHN, 10k samples out of the train set are reserved for validation and 40k for CIFAR10. As to Celeba, we use the official validation set for validation. The model that is kept at the end of training is the one achieving the best validation loss. All the models are trained with a batch size of 100 and starting learning rate of $1e-3$ (but CIFAR where the learning rate is set to $5e-4$) with an Adam optimizer (Kingma & Ba, 2014). We also use a scheduler decreasing the learning rate by half if the validation loss stops increasing for 5 epochs. For the experiments on the sensitivity to the training set size, we keep the same set-up. For each dataset we ensure that the validation set is $1/5^{\text{th}}$ the size of the train set but for CIFAR where we select the best model on the train set. The neural networks architectures can be found in Table 3 and are inspired by Ghosh et al. (2020). The metrics (FID and PRD scores) are computed with 10000 samples against the test set (for Celeba we selected only the 10000 first samples of the official test set). The factor $\rho$ is set to $\rho = \max_i \min_{j \neq i} \| c_i - c_j \|_2$ to ensure some *smoothness* of the manifold. For models coming from peers, we use the parameters provided by the authors when available.

For the data augmentation task, the generative models are trained on each class for 1000 epochs with a batch size of 100 and a starting learning rate of $1e-4$. Again a scheduler is used and the learning rate is cut by half if the loss does not improve for 20 epochs. All the models have the autoencoding architecture described in Table 3. As to the classifier, it is trained with a batch size of 200 for 50 epochs with a starting learning rate of $1e-4$ and Adam optimizer. A scheduler reducing the learning rate by half every 5 epochs if the validation loss does not improve is again used. The best kept model is the on achieving the best balanced accuracy on the validation set. Its neural network architecture may be found in Table 4. MRIs are only pre-processed such that the maximum value of a voxel is 1 and the minimum 0 for each data point.

Table 3: Neural networks used for the encoder and decoders of VAEs in the benchmarks

| | MNIST [CIFAR10] | SVHN | CELEBA | OASIS |
|---|---|---|---|---|
| ENCODER | (1[3], 32, 32) | (3, 32, 32) | (3, 64, 64) | (1, 208, 176) |
| LAYER 1 | CONV(128, (4, 4), STRIDE=2) BATCH NORMALIZATION RELU | LINEAR(1000) RELU | CONV(128, (5, 5), STRIDE=2) BATCH NORMALIZATION RELU | CONV(64, (5, 5), STRIDE=2) RELU |
| LAYER 2 | CONV(256, (4, 4), STRIDE=2) BATCH NORMALIZATION RELU | LINEAR(500) RELU | CONV(256, (5, 5), STRIDE=2) BATCH NORMALIZATION RELU | CONV(128, (5, 5), STRIDE=2) RELU |
| LAYER 3 | CONV(512, (4, 4), STRIDE=2) BATCH NORMALIZATION RELU | LINEAR(500, 16*) | CONV(512, (5, 5), STRIDE=2) BATCH NORMALIZATION RELU | CONV(256, (5, 5), STRIDE=2) RELU |
| LAYER 4 | CONV(1024, (4, 4), STRIDE=2) BATCH NORMALIZATION RELU | - | CONV(1024, (5, 5), STRIDE=2) BATCH NORMALIZATION RELU | CONV(512, (5, 5), STRIDE=2) RELU |
| LAYER 5 | LINEAR(4096, 16*) | - | LINEAR(16384, 64*) | CONV(1024, (5, 5), STRIDE=2) RELU |
| LAYER 6 | - | - | - | LINEAR(4096, 16*) |
| DECODER | (16 [32]) | (16) | (64) | (16) |
| LAYER 1 | LINEAR(65536) RESHAPE(1024, 8, 8) | LINEAR(500) RELU | LINEAR(65536) RESHAPE(1024, 8, 8) | LINEAR(65536) RESHAPE(1024, 8, 8) |
| LAYER 2 | CONVT(512, (4, 4), STRIDE=2) BATCH NORMALIZATION RELU | LINEAR (1000) RELU | CONVT(512, (5, 5), STRIDE=2) BATCH NORMALIZATION RELU | CONVT(512, (5, 5), STRIDE=(3, 2)) RELU |
| LAYER 3 | CONVT(256, (4, 4), STRIDE=2) BATCH NORMALIZATION RELU | LINEAR(3072) RESHAPE(3, 32, 32) SIGMOID | CONVT(256, (5, 5), STRIDE=2) BATCH NORMALIZATION RELU | CONVT(256, (5, 5), STRIDE=2) RELU |
| LAYER 4 | CONVT(3, (4, 4), STRIDE=1) BATCH NORMALIZATION SIGMOID | - | CONVT(128, (5, 5), STRIDE=2) BATCH NORMALIZATION RELU | CONVT(128, (5, 5), STRIDE=2) RELU |
| LAYER 5 | - | - | CONVT(3, (5, 5), STRIDE=1) BATCH NORMALIZATION SIGMOID | CONVT(64, (5, 5), STRIDE=2) RELU |
| LAYER 6 | - | - | - | CONVT(1, (5, 5), STRIDE=1) RELU |

[2]MNIST images are re-scaled to 32x32 images with a 0 padding.

Table 4: Neural Network used for the classifier in Sec. 5.3

| | OASIS CLASSIFIER |
|---|---|
| INPUT SHAPE | (1, 208, 176) |
| LAYER 1 | CONV(8, (3, 3), STRIDE=1)
BATCH NORMALIZATION
LEAKYRELU
MAXPOOL(2, STRIDE=2) |
| LAYER 2 | CONV(16, (3, 3), STRIDE=1)
BATCH NORMALIZATION
LEAKYRELU
MAXPOOL(2, STRIDE=2) |
| LAYER 3 | CONV(32, (3, 3), STRIDE=2)
BATCH NORMALIZATION
LEAKYRELU
MAXPOOL(2, STRIDE=2) |
| LAYER 4 | CONV(64, (3, 3), STRIDE=2)
BATCH NORMALIZATION
LEAKYRELU
MAXPOOL(2, STRIDE=2) |
| LAYER 5 | LINEAR(256, 100)
RELU |
| LAYER 6 | LINEAR(100, 2)
SOFTMAX |

# F    DATASET SIZE SENSIBILITY ON SVHN

In Figure 10, we show the same plot for SVHN as in Sec. 5.2. Again the proposed method appears to be part of the most robust generation procedures to dataset size changes.

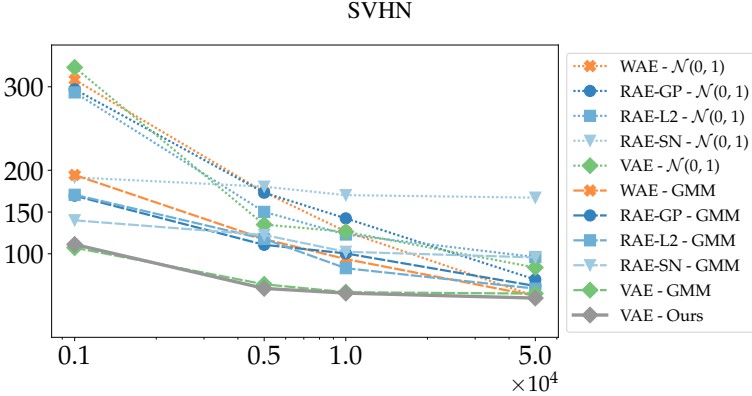

Figure 10: FID score evolution according to the number of training samples.

# G ABLATION STUDY

## G.1 INFLUENCE OF THE NUMBER OF CENTROIDS IN THE METRIC

In order to assess the influence of the number of centroids and their choice in the metric in Eq. 7, we show in Figure 11 the evolution of the FID according to the number of centroids in the metric (left) and the variation of FID according to the choice in the centroids. As expected, choosing a small number of centroids will increase the value of the FID since it reduces the variability of the generated samples that will remain *close* to the centroids. Nonetheless, as soon as the number of centroids is higher than 1000 the FID score is either competitive or better than peers and continues decreasing as the number of centroids increases.

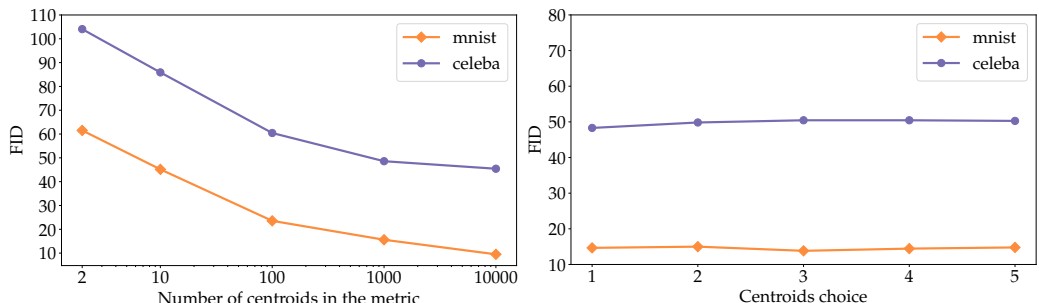

Figure 11: *Left:* FID score evolution according to the number of centroids in the metric (Eq. 7). *Right*: The FID variation with respect to the choice in centroids. We generate 10000 samples by selecting each time different centroids ($k = 1000$).

To assess the variability of the generated samples, we propose to analyze some generated samples when only 2 centroids are considered. In Figure 12, we display on the left the decoded centroids along with the closest image to these decoded centroids in the train set. On the right are presented some generated samples. We place these samples in the top row if they are closer to the first decoded centroid and in the bottom row otherwise. Interestingly, even with a small number of centroids the proposed sampling scheme is able to access to a relatively good diversity of samples. These samples are not simply resampled train images or a simple interpolation between selected centroids as some of the generated samples have attributes such as glasses that are not present in the images of the decoded centroids.

Decoded centroid  Nearest train image               Generated samples

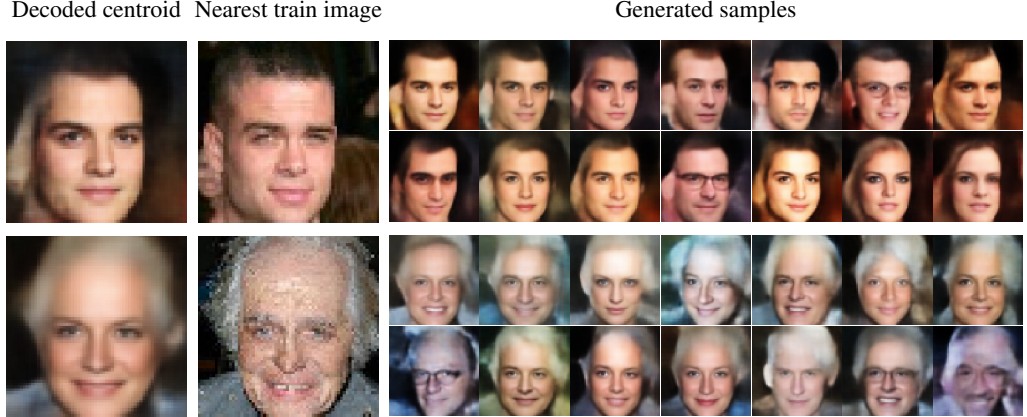

Figure 12: Variability of the generated samples when only two centroids are considered in the metric. *Left:* The image obtained by decoding the centroids. *Middle*: The nearest image in the train set to the decoded centroids. *Right:* Some generated samples. Each generated sample is assigned to the closest decoded centroid (top row for the first centroid and bottom row for the second one).

## G.2  INFLUENCE OF $\lambda$ IN THE METRIC

In this section, we also assess the influence of the regularization factor $\lambda$ in Eq. 7 on the resulting sampling. To do so, we generate 10k samples using the proposed method on both MNIST and Celeba datasets for values of $\lambda \in [1e^{-6}, 1e^{-4}, 1e^{-2}, 1e^{-1}, 1]$. Then, we compute the FID against the test set. Each time, we consider $k = 1000$ centroids in the metric. As shown in Figure 13, the influence of $\lambda$ remains limited. In practice, $\lambda$ is mainly here to avoid pathological cases such as the metric collapsing to **0** *far* from the centroids. In the implementation, a typical choice for $\lambda$ is $1e^{-2}$.

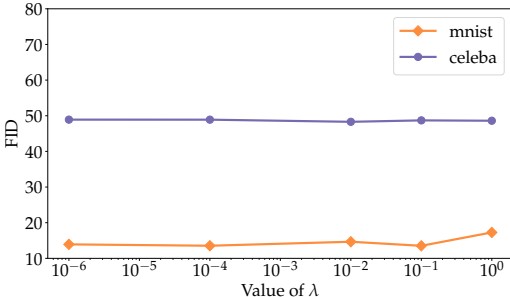

Figure 13: FID score evolution according to the value of $\lambda$ in the metric (Eq.7).

## G.3  THE CHOICE OF $\rho$

In the experiments presented, the smoothing factor $\rho$ in Eq. 7 is set to the value of the maximum distance between two closest centroids $\rho = \max_{i} \min_{j \neq i} \|c_j - c_i\|_2$. This choice is motivated by the fact that we wanted to build a smooth metric and so ensure some *smoothness* of the manifold while trying to interpolate faithfully between the metric tensors $\mathbf{G}_i = \boldsymbol{\Sigma}_i^{-1}$. In particular, a small value of $\rho$ would have allowed disconnected regions and the sampling may have not prospected well the learned manifold and would have only become a resampling of the centroids. On the other hand, setting a high value for $\rho$ would have biased the interpolation and the value of the metric at a $\boldsymbol{\mu}_i$. As a result, $\mathbf{G}(\boldsymbol{\mu}_i)$ might have been very different from the one observed $\boldsymbol{\Sigma}_i^{-1}$ since the other $\boldsymbol{\mu}_j$ would have had a strong influence on its value. The proposed value for $\rho$ appeared to work well in practice.

# H    OTHER CLASSIFICATION METRICS

In Table 5 are presented some further classification results for each considered model while generated samples using each generation procedure are made available in Figure 14. On OASIS database the proposed method appears to produce visually the sharpest samples. This better generation performance is also supported by the classification metrics provided in Table 2 and Table 5.

Table 5: Classification results averaged on 20 independent runs. For the VAEs, the classifier is trained on 2K generated samples per class.

| Generation method | Balanced Accuracy | Precision | | Recall | |
|---|---|---|---|---|---|
| | | AD | CN | AD | CN |
| Original* | $66.2 \pm 7.6$ | $74.7 \pm 8.4$ | $80.3 \pm 4.0$ | $35.7 \pm 16.3$ | $95.7 \pm 1.5$ |
| Original (resampled) | $81.8 \pm 2.6$ | $67.0 \pm 5.3$ | $91.4 \pm 1.8$ | $78.5 \pm 5.2$ | $85.1 \pm 4.2$ |
| AE - $\mathcal{N}$ | $50.0 \pm 0.0$ | $0.0 \pm 0.0$ | $72.6 \pm 0.0$ | $0.0 \pm 0.0$ | $100.0 \pm 0.0$ |
| WAE | $57.4 \pm 9.7$ | $48.5 \pm 42.8$ | $76.7 \pm 6.1$ | $19.3 \pm 27.5$ | $95.4 \pm 9.3$ |
| VAE - $\mathcal{N}$ | $51.8 \pm 3.8$ | $38.0 \pm 47.3$ | $73.4 \pm 1.7$ | $3.7 \pm 7.8$ | $99.8 \pm 0.7$ |
| VAMP | $83.1 \pm 2.6$ | $56.3 \pm 5.2$ | $97.5 \pm 2.1$ | $94.8 \pm 4.7$ | $71.5 \pm 7.4$ |
| HVAE | $56.3 \pm 7.9$ | $48.7 \pm 41.7$ | $75.5 \pm 3.8$ | $13.9 \pm 17.6$ | $98.6 \pm 2.2$ |
| RHVAE | $68.0 \pm 10.9$ | $56.1 \pm 25.3$ | $83.0 \pm 7.5$ | $46.7 \pm 30.2$ | $89.2 \pm 10.6$ |
| AE - GMM | $82.4 \pm 2.3$ | $55.8 \pm 4.9$ | $96.8 \pm 2.4$ | $93.3 \pm 5.6$ | $71.5 \pm 6.2$ |
| RAE (GP) | $63.9 \pm 9.8$ | $45.3 \pm 18.5$ | $84.2 \pm 8.6$ | $60.9 \pm 28.6$ | $67.0 \pm 24.9$ |
| RAE (L2) | $74.1 \pm 6.0$ | $57.8 \pm 10.1$ | $88.3 \pm 5.2$ | $70.0 \pm 18.7$ | $78.3 \pm 11.7$ |
| RAE (SN) | $62.3 \pm 8.9$ | $43.1 \pm 24.9$ | $80.6 \pm 6.6$ | $41.7 \pm 30.1$ | $82.9 \pm 16.4$ |
| RAE | $69.3 \pm 8.1$ | $56.2 \pm 13.5$ | $85.2 \pm 6.2$ | $60.0 \pm 24.0$ | $78.5 \pm 17.5$ |
| VAE - GMM | $83.0 \pm 3.6$ | $60.7 \pm 5.4$ | $94.9 \pm 3.7$ | $88.0 \pm 9.5$ | $77.9 \pm 5.9$ |
| VAE - Ours | $85.4 \pm 2.5$ | $64.0 \pm 5.3$ | $95.8 \pm 2.2$ | $90.4 \pm 5.6$ | $80.3 \pm 5.1$ |

*unbalanced

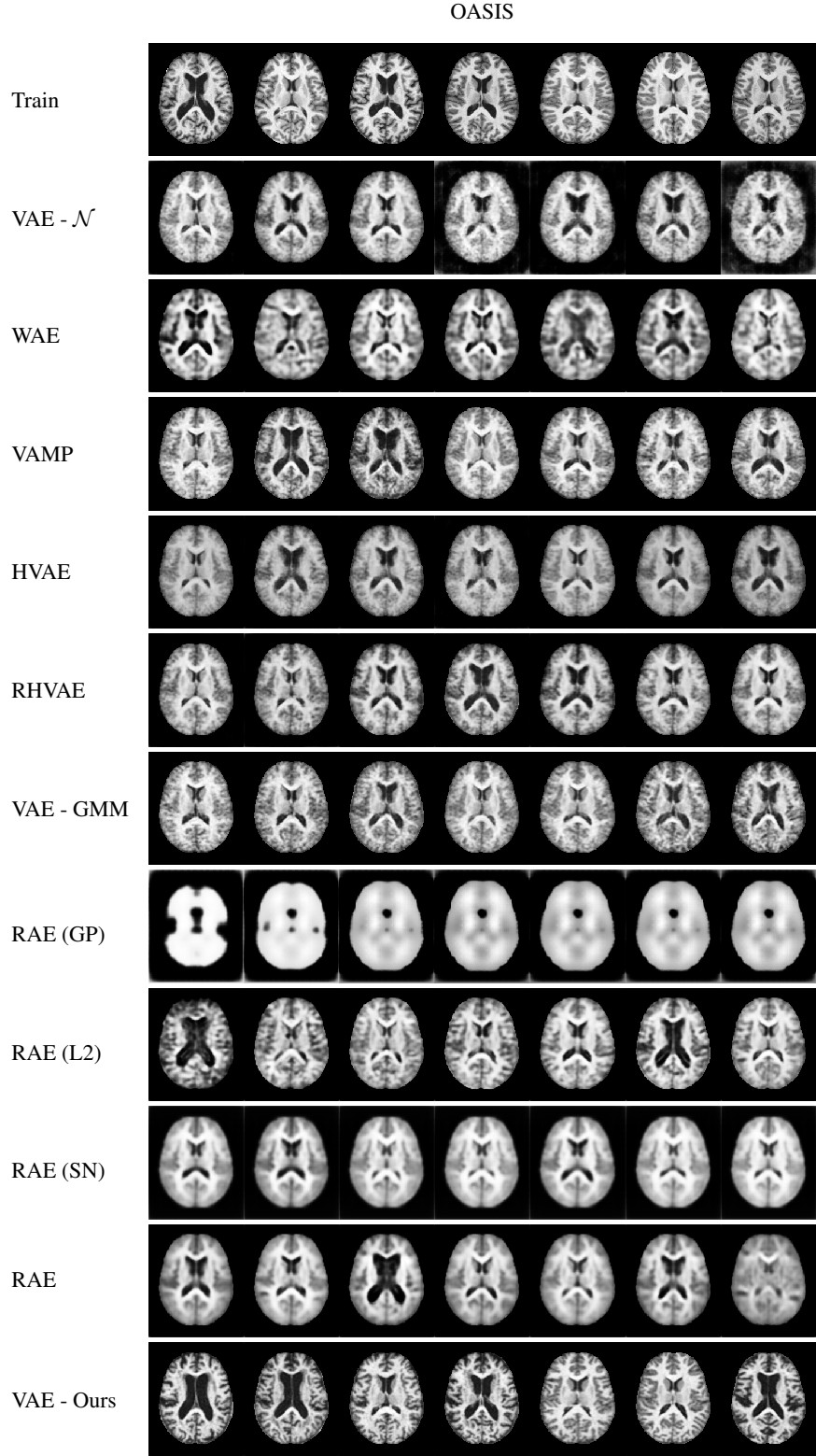

Figure 14: Generated samples with different models and generation processes.

# I    LINK BETWEEN THE RIEMANNIAN VAE AND VANILLA VAE

We assume as in (Ghosh et al., 2020) that a VAE is essentially an autoencoder regularized with noise. Hence, a Riemannian-based VAE could also be seen as a regularized autoencoder but the noise would be informed by the intrinsic geometry of the latent space. Indeed, the main assumption behind the Riemannian VAE would consist in assuming that given a set of data $\boldsymbol{x} \in \mathcal{X} \subset \mathbb{R}^D$ there exists a lower dimensional space, namely the latent space, that has apparently no reason to be Euclidean in which live the latent variables. In such a framework, it would be assumed that this space is the $d$-dimensional Riemannian manifold $\mathcal{M} = (\mathbb{R}^d, \mathbf{G})$ where $\mathbf{G}$ is an unknown Riemannian metric. Now the goal of the Riemannian VAE would be the same as a regularized autoencoder that is to learn a *smooth* representation of the data within a much lower dimensional space here seen as the Riemannian manifold $\mathcal{M}$.

To to so and similarly to autoencoder models, it would be assumed that there exist $e_\phi : \mathbb{R}^D \to \mathcal{M}$ a parametrized encoding function mapping the input data onto the manifold $e_\phi(\boldsymbol{x}) = \boldsymbol{\mu} \in \mathcal{M}$ and $d_\theta : \mathcal{M} \to \mathbb{R}^D$ a parametrized decoding function that maps back the latent codes to the data space. In such a case, the main objective would be to find $\phi$ and $\theta$ such that the reconstruction loss is minimized

$$\min_{\phi,\theta} \mathcal{L}_{\text{REC}} = \min_{\phi,\theta} l(\boldsymbol{x}, d_\theta(e_\phi(\boldsymbol{x})), \qquad x \in \mathcal{X}, \tag{14}$$

where $l$ is a function measuring the distance between the input data and the reconstructions and is chosen depending on the problem and the data (*e.g.* mean square error, binary cross entropy...). In order to learn a *smooth* latent space meaning that small variations in the latent space do not change completely the output of the decoder, the decoder is also regularized using a Riemannian Gaussian noise. That is imposing that

$$d_\phi(\boldsymbol{z}) \approx \boldsymbol{x}, \qquad \boldsymbol{z} \sim \mathcal{N}_{\text{riem}}^{\mathbf{G}}(\boldsymbol{z}|\boldsymbol{\mu}, \sigma),$$

where

$$\mathcal{N}_{\text{riem}}^{\mathbf{G}}(\mathbf{z}|\sigma, \boldsymbol{\mu}) = \frac{1}{C} \exp\Big(-\frac{\text{dist}_{\mathbf{G}}(\mathbf{z}, \boldsymbol{\mu})^2}{2\sigma}\Big), \quad C = \int_{\mathcal{M}} \exp\Big(-\frac{\text{dist}_{\mathbf{G}}(\boldsymbol{z}, \boldsymbol{\mu})^2}{2\sigma}\Big) d\mathcal{M}_{\boldsymbol{z}}, \tag{15}$$

and the reconstruction loss in Eq. 14 would become

$$\min_{\phi,\theta} \mathcal{L}_{\text{REC}} = \min_{\phi,\theta} l(\boldsymbol{x}, d_\theta(\boldsymbol{z})), \qquad x \in \mathcal{X}, \qquad \boldsymbol{z} \sim \mathcal{N}_{\text{riem}}^{\mathbf{G}}(\boldsymbol{z}|e_\phi(\boldsymbol{x}), \sigma),$$

Hence, we no longer decode the embedding $\boldsymbol{\mu}$ but rather $\boldsymbol{z}$ that is obtained with the Riemannian Gaussian distribution centered on $\boldsymbol{\mu}$. Since the metric $\mathbf{G}$ is unknown, a Riemannian VAE would aim at learning the metric directly from the data. Thus, the encoder function would output an embedding $\boldsymbol{\mu}$ of an input data point but also the value of the Riemannian metric at the embedding point *i.e.* $\mathbf{G}(\boldsymbol{\mu})$. Since, we would only have access to a finite number of metric tensors, a smooth metric $\mathbf{G}$ could be built using Eq. 7. Now, at least theoretically, we would be able to compute the geodesic distance involved in Eq. 15. As of now, the manifold is not regularized and so pathological cases such as the metric collapsing to $0$ may arise. To avoid such a behavior, some smoothness conditions could be applied on the manifold by imposing for instance that the Riemannian Gaussian distribution is not *too far* from a standard Gaussian. This would prevent the metric from collapsing to $0$ and ensure that the latent codes remain close to the origin as well. However, other regularization schemes could have be envisioned as well. Since we would be working in an *ambient-like* manifold, there exists a global chart $z$ and so the density of the Riemannian Gaussian distribution can be written with respect to the Lebesgue measure $d\boldsymbol{z}$ in $\mathbb{R}^d$ (Pennec, 2006)

$$\mathcal{N}_{\text{riem}}^{\mathbf{G}}(\mathbf{z}|\sigma, \boldsymbol{\mu}) = \frac{1}{C} \exp\Big(-\frac{\text{dist}_{\mathbf{G}}(\mathbf{z}, \boldsymbol{\mu})^2}{2\sigma}\Big) \sqrt{\det \mathbf{G}(\boldsymbol{z})}$$

$$C = \int_{\mathbb{R}^d} \exp\Big(-\frac{\text{dist}_{\mathbf{G}}(\boldsymbol{z}, \boldsymbol{\mu})^2}{2\sigma}\Big) \sqrt{\det \mathbf{G}(\boldsymbol{z})}\, d\boldsymbol{z}, \tag{16}$$

Hence, the regularization term set as the KL divergence between the Riemannian Gaussian distribution and the standard normal would follow

$$\mathcal{L}_{\text{REG}} = D_{\text{KL}}\Big(\mathcal{N}_{\text{riem}}^{\mathbf{G}}(\mathbf{z}|\sigma, \boldsymbol{\mu}) \| \mathcal{N}(0, I_d)\Big)$$

$$= \int_{\mathbb{R}^d} p_{\mathcal{N}_{\text{riem}}^{\mathbf{G}}(\sigma,\boldsymbol{\mu})}\Big(\log(p_{\mathcal{N}_{\text{riem}}^{\mathbf{G}}(\sigma,\boldsymbol{\mu})}) - \log(p_{\mathcal{N}(0,I_d)})\Big) d\boldsymbol{z}\,.$$

The final objective a Riemannian VAE would try to minimize would then write

$$\mathcal{L} = \mathcal{L}_{\text{REC}} + \mathcal{L}_{\text{REG}}\,.$$

Unfortunately, this framework cannot be used in practice for at least two reasons. The first one is the sampling from the Riemaniann distribution in Eq. 15 which is far from being trivial. MCMC methods could have been envisioned to sample from such a distribution but this would have impeded backpropagation since this framework would not be amenable to the reparametrization trick (Salimans et al., 2015; Caterini et al., 2018). Second, the regularization term would involve computing the density of the Riemannian Gaussian distribution which explicitly involves the computation of the Riemannian distance and so the resolution of the optimization problem in Eq. 9. Since, this framework is not usable in practice, it can be assumed that the value of the metric during training can be approximated by its value at $\boldsymbol{\mu}$ (*i.e.* $\mathbf{G}(\boldsymbol{\mu})$). With this approximation, Riemannian Gaussians become multivariate Gaussians and all the terms become computable. We find back the vanilla VAE framework if we further consider $\mathbf{G}(\boldsymbol{\mu}) = \boldsymbol{\Sigma}(\boldsymbol{x})^{-1}$ where $\boldsymbol{\Sigma}(\boldsymbol{x})$ is the covariance matrix given by the encoder of a vanilla VAE. We indeed have

$$
\begin{aligned}
\mathcal{N}_{\text{riem}}^{\boldsymbol{\Sigma}^{-1}}(\mathbf{z}|\sigma, \boldsymbol{\mu}) &= \frac{1}{C} \exp\Big( -\frac{\text{dist}_{\boldsymbol{\Sigma}^{-1}}(\mathbf{z}, \boldsymbol{\mu})^2}{2\sigma} \Big) \sqrt{\det \boldsymbol{\Sigma}^{-1}} \\
C &= \int_{\mathbb{R}^d} \exp\Big( -\frac{\text{dist}_{\boldsymbol{\Sigma}^{-1}}(\boldsymbol{z}, \boldsymbol{\mu})^2}{2\sigma} \Big) \sqrt{\det \boldsymbol{\Sigma}^{-1}}\, d\boldsymbol{z}\,,
\end{aligned}
\tag{17}
$$

where $\text{dist}_{\boldsymbol{\Sigma}^{-1}}(\boldsymbol{z}, \boldsymbol{\mu}) = \sqrt{(\boldsymbol{z} - \boldsymbol{\mu})^\top \boldsymbol{\Sigma}^{-1}(\boldsymbol{z} - \boldsymbol{\mu})}$. If we further set $\sigma = 1$ we have

$$
\mathcal{N}_{\text{riem}}^{\boldsymbol{\Sigma}^{-1}}(\mathbf{z}|\sigma, \boldsymbol{\mu}) = \frac{\exp\Big( -\frac{(\boldsymbol{z}-\boldsymbol{\mu})^\top \boldsymbol{\Sigma}^{-1}(\boldsymbol{z}-\boldsymbol{\mu})}{2} \Big)}{\int_{\mathbb{R}^d} \exp\Big( -\frac{(\boldsymbol{z}-\boldsymbol{\mu})^\top \boldsymbol{\Sigma}^{-1}(\boldsymbol{z}-\boldsymbol{\mu})}{2} \Big)\, d\boldsymbol{z}}\,.
\tag{18}
$$

Assuming as is standard in the VAE framework, that $\boldsymbol{\Sigma}$ is diagonal makes the computation of $\mathcal{L}_{\text{REG}}$ easy and we retrieve the training of a vanilla VAE model. Indeed, as explained in Remark. 1, the log of the conditional distribution $p_\theta$ reduces to the reconstruction loss $\mathcal{L}_{\text{REC}}$ and the KL between the variational posterior $q_\phi(z|x)$ and the prior taken as a standard Gaussian gives $\mathcal{L}_{\text{REG}}$.

