# OpenReview forum: "A Geometric Perspective on Variational Autoencoders"
_ICLR.cc/2022/Conference — ICLR 2022 Submitted_

### Official Review · Reviewer_v2S3 · 2021-10-27

**Correctness:** 2
**Technical Novelty And Significance:** 3
**Empirical Novelty And Significance:** 2
**Recommendation:** 6
**Confidence:** 4

**Main Review:**

Strengths:
- The proposed approach, i.e., using the Riemannian geometry to formulate distributions in the VAE, is interesting.
- The paper proposes a general perspective on VAEs through a geometrical perspective. It generalizes some papers on VAEs with various latent manifolds (e.g., hyperspherical VAE, VAEs with Poincare disk).

Weaknesses:
- Remark 1 is not necessarily true for two reasons. First, this is not true that $p_{\theta}(\mathbf{x}|\mathbf{z})$ is often taken as a multivariate Gaussian. The conditional distribution must be chosen according to the problem at hand. If data is binary, the Bernoulli distribution is taken; if data is continuous, we can pick from a plethora of distributions. Second, a very common choice to model RGB images is a discretized logistic distribution rather than a Gaussian distribution.
- The example in Sec. 4.5 is a bit vague. First, the setup is not well explained. Second, Figure 1 is not properly explained: there are three times (a), and one (b), but they are neither referred to in the caption nor in the text. Moreover, the font used for the top (a) and top (b) are so small that it is hard to read.
- The paper explains the idea, however, it lacks a more concrete statement of the proposed approach. The authors mention that they use Eq. (7) as the Riemannian metric and the HMC sampler for sampling, however, it is not necessarily obvious what they precisely use later on in the experiments. It would be highly beneficial to provide a specific instantiation of their approach, e.g., how the metric is calculated, what is the final objective, what is the sampling procedure, what is the training algorithm. At the moment, the paper presents the ideas at a rather general level.



**Summary Of The Paper:**

The paper proposes to generalize the vanilla VAE by formulating the conditional distribution and the marginal distribution using the Riemannian geometry. The authors proposed a generalized perspective on various VAEs proposed so far. The paper is rather easy to follow, however, it is unclear what precisely is used in the experiments and how the proposed approach works.

**Summary Of The Review:**

I like the paper and the idea of using Riemannian geometry in VAEs. However, the paper presents the idea at a pretty general level and it is hard to see the details that matter from the computational perspective. I doubt it would be easy and straightforward to code the proposed approach up. This is my main problem with the paper, namely, the reproducibility. And the follow-up problem is about the computational burden of the proposed approach. Since it is unclear how to sample and how to calculate various components of the VAE, it is hard to assess how “heavy” the method is and whether it could be scaled up. As we see nowadays, models that are easy to scale up (e.g., hierarchical VAEs like NVAE or diffusion-based models) allow achieving SOTA performance.

=== UPDATE ===
I appreciate the rebuttal. Overall, I still doubt whether the paper is a strong contribution. Nevertheless, the new additions (especially Appendix C) make the paper better. Therefore, I decided to increase my score to 6.

---

> ### Author Response · Authors · 2021-11-17
> **Thank you for your review!**
>
> Dear reviewer,
>
> Thank you for your time and your comprehensive review of our paper. Thank you also for liking the idea described in the paper. We will try to address your concerns below.
>
> ### On remark 1
> You are right, these exist many choices for the decoding distribution depending on the input data we are dealing with. The aim of this remark was to provide an example when the decoding distribution is Gaussian. However, we agree that was misleading. Hence, we have amended it in the revised manuscript to better reflect the variability in distributions that may be chosen in the decoder.
>
> ### On example 4.5
> We changed the caption so that it better explains the results. Thank for spotting the typo, this was rectified in the updated version.
>
> ### Lack of concrete statement.
> We understand that the paper presents the method at a rather general level. To make the method more reproducible we have added in Appendix.C a comprehensive explanation on how to use the method in practice along with pseudo-code algorithms to better reflect the complexity of the proposed scheme. In particular, it explains how to recover the metric from a trained VAE model, clarifies that the training process of the VAE remains unchanged, shows the different steps of the sampling procedure and describes the distribution and elements that intervene within the HMC sampler and how to compute them. The comprehensive experimental set-up can be found in Appendix E. We also added a reproducibility statement at the end of the paper mentioning the relevant sections of the paper allowing for reproducibility. Please note that this does not count toward the 9 page limit as per ICLR guidelines.
>
> Please do not hesitate if you have any further questions/comments.

---

> > ### Comment · Reviewer_v2S3 · 2021-11-24
> > **I increase my score to 6**
> >
> > Dear authors,
> >
> > Thank you for your rebuttal. I really appreciate your hard work!
> > I share some doubts with other reviewers (e.g., the impact of this paper could be limited due to its general statements but, on the other hand, some statements are very limiting, i.e., Gaussianity). Nevertheless, I find the new Appendix C as a good way to increase reproducibility. Therefore, I decided to increase my score to 6.
> >
> > Best!

---

### Official Review · Reviewer_BQpw · 2021-10-29

**Correctness:** 3
**Technical Novelty And Significance:** 4
**Empirical Novelty And Significance:** 3
**Recommendation:** 6
**Confidence:** 3

**Main Review:**

This work proposes a novel interpretation of variational encoding by the prism of differential geometry.  In particular the data $x_i$ provides two main elements, the “position” $\mu(x_i)$ and instead of interpreting the estimated $\Sigma(x_i)$ as the variance of the normal law i.e of p(z|x) it is interpreted as the value of the Riemanian metric at $\mu(x_i)$. In that perspective, the authors propose to sample not from an isotropic Gaussian of $\mathbb{R}^d$, but rather using uniformly from the learned manifold.
The experiments are interesting and well-conducted.

Remarks and Question:
1. On the interpretation of the Geometric interpretation of the VAE. Section 4.3 is a little confusing. Indeed, the initial assumption that the inverse variance corresponds to the local metric tensor is acceptable (the lower the variance the less the sample is allowed to move from the mean). However, the rest of the section is in my opinion confusing. Indeed, because the authors seem to suggest that the original-VAE is in fact an approximation of a Riemannian VAE, can the authors link the Riemannian VAE penalties and how they are approximated in the original VAE framework? For clarity, I highly recommend including such discussion in the final version.
2. Can the authors comment on the link between their work and the desire of the VAE-community to rather enrich the expressivity of p(z|x) using normalizing flows or Hamiltonian scheme [1, 2]
3. I can’t relate eq.7 to the interpretation of the inverse variance as a local Riemannian metric (i.e $G(\mu(x_i) = \Sigma^{-1}(x_i)$). Indeed as of eq.7 $G(\mu(x_i)) \neq \Sigma^{-1}$, depending on how $\mu(x_i)$ is close to the other $\mu_i$.
4. In the same line can the author comment on the need for the regularization term $+ \lambda I_d$ ? How to choose $\lambda$ in practice ?

On the Experiments:
1. What is the influence of the number of selected points in the sampling based on 7). For example can the author provide a FID curve somehow like Fig.4 with varying number of “support vectors” $(\mu_i, \Sigma^{-1}_i)$ ? Indeed, the more complex the latent space geometry, the more “support-vectors” it may require for an acceptable decoding.
2. Because the sampling method in the latent space is dependent on the chosen support vectors, did the authors observe any variance in the quality of the samples depending on the choice of the K-centroids (something like variance of the FID at a defined number of support vectors)?
3. On the Decoding processes : One drawback of such sampling scheme (due to the choice of the centroids) is the fact that in the decoding processes, the decoder simply provides an interpolation between the decoded K-support vectors. Did the authors observe such an issue in practice ?

I would be glad to increase my score if the major concerns are answered.

[1]: Hamiltonian Variational Auto-Encoder,  Anthony L. Caterini, Arnaud Doucet, Dino Sejdinovic

[2]: Variational Inference with Normalizing Flows, Danilo Jimenez Rezende, Shakir Mohamed


**Summary Of The Paper:**

The paper proposes to analyse the VAE framework using differential geometry tools. More specifically, they propose to impose a (Riemannian) metric in the latent space of a VAE. Such a Riemanian metric then enables data generation based on the (Riemanian) Uniform distribution on the learned latent space, improving the quality of the generated samples.


**Summary Of The Review:**

The geometric interpretation of the VAE is interesting. Moreover, the author's proposition for resampling seems sound in the light of their proposition and the experiments are well conducted. Yet, the theoretical link between the VAE and the Riemannian distance based VAE requires a more thorough discussion.

---

> ### Author Response · Authors · 2021-11-17
> **Thank you for your review!**
>
> Dear reviewer,
>
> Thank you for your time and efforts reviewing our paper. Thank you also for your comprehensive review of the paper and finding our work interesting.
>
> ### On the geometric interpretation
> We have added a short -because of page limitations- paragraph at the end of section 4.3 to clarify the limitations of a fully Riemannian model and the necessity of the approximations performed by the vanilla VAE in particular during training.
>
> ### Normalzing Flows and HVAE
> Thank you for the references, we indeed think that methods aiming at improving the posterior distribution are relevant to this paper as well since this is another interesting approach to improve the VAE model that has been widely discussed in the literature. Hence, we have added some discussion about those methods in the "related work" section of the revised manuscript. For completeness, we also re-implemented and performed the same experiments with a Hamiltonian VAE.
>
> ### On Eq.7
> This is true that the value of the metric obtained with Eq.7 may differ slightly from $\Sigma_i^{-1}$ at encoded means. This is due to the proposed interpolation method that is needed to build a smooth continuous metric. Nonetheless, the choice of $\rho$ as the maximum distance between two closest centroids combined with an exponential kernel aimed at reducing the impact of the other $\mu_j$ while ensuring some smoothness of the manifold (i.e. ensuring that the metric varies smoothly).
>
> ### On $\lambda$
> The introduction of a regularization term is mainly for computational purposes and pathological cases such as the metric collapsing to 0 *far* from the centroids. Theoretically, such regularization may not be needed since the form of the metric provided with Eq. 7 always ensure that G is symmetric positive definite. We conducted in Appendix. G an ablation study to assess the impact of $\lambda$ on the proposed sampling quality.
>
> ### Influence of the number of centroids
> Thank you for raising this question. The number of centroids has indeed an impact on the resulting sampling. The more centroids we have the more expressive the sampling should be. We added in Appendix. G the evolution of the FID with respect to the number of centroids in the metric for two datasets (MNIST & Celeba)
>
> ### The impact of the choice of the centroids
> We also discussed this aspect in Appendix.G where we plot the variation of the FID for 5 different choices in the centroids ($k=1000$) for two datasets (MNIST & Celeba)
>
> ### On the decoding process
> That is true that the sampling is biased toward the chosen centroids. Nonetheless, the shape of the metric (and so of the distribution we propose to sample from) allows, thanks to $\rho$, to prospect the manifold around the centroids and not only *between* them. Hence, there is no reason that the decoded samples are a simple interpolation of the decoded centroids. We also illustrate that aspect in Appendix. G where we generate data on Celeba with only 2 centroids in the metric. The sampling allowed to generate images that indeed look like the decoded centroids but also have attributes that are not present in any of the decoded centroids (e.g. glasses).
>
> Please do not hesitate if you have any further questions/comments.

---

> > ### Comment · Reviewer_BQpw · 2021-11-19
> > **Regarding authors response**
> >
> > Thanks to the authors for providing novel experimental results and explanations on the method, improving in my opinion the overall quality of the paper.
> >
> > ### Regarding Novel Experimental Results
> > - In particular the results of appendix G. are particularly interesting and shows that despite the sampling is biased, the proposed sampling scheme manages somehow to explore at least locally the data manifold, showing a variety in the generated gender, glasses, and hairs (but less in the posture).
> >
> > - Also the FID suffer a low variance regarding the chosen centroids (as soon as it is high).
> > Indeed, I doubt that similar results could be obtained with lower values of k (e.g 10 centroids).
> >
> > ### Regarding the Riemann VAE motivation
> > The discussion at the end of section 4.3 brings substance and come to support the method.
> >
> >
> > ### Final Recommendations and Remarks
> > - As a final recommendation, I suggest the authors a thorougher derivation of the Riemann based VAE in the appendices for a better understanding of the shortcomings in training such a method.
> > - Despite pseudo-code being available in the Appendices, I highlight recommand the publication of the code.
> >
> > Despite the novelty in the contributions not being very high, it definitely follows an interesting research path in generative modeling to refine the sampling method, different from posterior enrichment.
> >
> > The well conducted experimental section makes this proposition useful for practitioner. The weakest part of the proposition is the lack of theoretical investigation linking the Riemann based VAE and the original one.
> >
> > Therefore, I believe this paper is borderline, but its practical impact can be interesting. In that perspective, the merits may overcome the limitations, thus, I increase my score to 6.

---

> > > ### Author Response · Authors · 2021-11-20
> > > **Thank you for your reply!**
> > >
> > > Dear Reviewer,
> > >
> > > We thank you for your prompt reply and your positive feedback regarding the additional experiments and explanations we have added to the manuscript. We are glad to see that they have either clarified or better supported some of our claims.
> > >
> > > ### Riemannian VAE
> > > We appreciate the concerns of the reviewer regarding the link between the Riemannian VAE and the vanilla one. Further to your last comment we have uploaded a new version of the manuscript with a new Appendix (Appendix I) that derives more precisely what could have been the Riemannian VAE, details what would be the main shortcomings in training this model and shows how the vanilla VAE version approximate such a model.
> > >
> > > ### Publication of the code
> > > We have provided an implementation of the method as supplementary material but the code will also be released on a public platform such as github if the paper is accepted.
> > >
> > > Please do not hesitate if you have any further comments/questions.

---

> > > > ### Author Response · Authors · 2021-12-01
> > > > **Gentle reminder**
> > > >
> > > > Dear reviewer,
> > > >
> > > > Thank you again for your review and comments following our rebuttal. As the end of the discussion period is approaching, we were wondering if you had time to look at our last addition concerning the link between the vanilla VAE and the Riemannian one in Appendix I?
> > > >
> > > > Do not hesitate if you have any further questions.
> > > >
> > > > Regards,
> > > >
> > > > The authors

---

### Official Review · Reviewer_yrKH · 2021-11-01

**Correctness:** 4
**Technical Novelty And Significance:** 2
**Empirical Novelty And Significance:** 2
**Recommendation:** 6
**Confidence:** 3

**Main Review:**


The idea is to equip the latent space with a Riemannian structure, the measure of which depends on the inverse of covariance of the posterior.
Before sampling, the distribution needs to be approximated using MCMC.
The results are ok.

Pros:
* The article is well written and smoothly walks the reader through the mathematics behind variational autoencoders (VAE) and Riemannian geometry.
* The idea is clear and straightforward. The theory is robust. However, I am doubtful of the practical outcomes.

Cons:
* The analyses present several shortcomings.
* The baselines are chosen accordingly to the theoretical line of the paper. That is to improve the samples' quality without touching (too much) the loss nor the architecture.
That is fair, although I was expecting some Energy-based approaches such as VAEBM (Xiao et al. 2020) or Pang et al. (which is in the references but not cited)
* The authors overemphasize sometimes.
** "Now that the VAE has learned and capture the intrinsic geometry of the data within the latent space seen as a Riemannian manifold,"
**    Why is the variance-based measure necessarily the one that captures the "intrinsic geometry of the data"?
    Besides, what if the data is made of several connected components? Such a situation is not allowed by the theory developed here.
**  VAE+GMM is almost as good as the proposed method in Table 2.


**Summary Of The Paper:**

It is a theoretical paper that discusses the Riemannian structure of the latent space of a VAE.
The associated measure on a point \mu(x) derives from \Sigma^-1(x), which are the parameters of the posterior p(z|x) or simply the outputs of the encoder.
The measure allows to define a Riemannian normal distribution on that space/manifold, to compute geodesics but also to "better" sample.

**Summary Of The Review:**

The document is good, easy to read, and fits within the scope of the conference.
The Riemannian point of view on VAE surely brings interesting insights that might lay the foundation of other works.
Nevertheless, I have doubts about the importance of the contributions and their impact on the field.
Moreover, the analysis is pretty basic. Maybe because there is too much to say/study for a conference paper.

---

> ### Author Response · Authors · 2021-11-17
> **Thank you for your review!**
>
> Dear reviewer,
>
> Thank you for taking the time to review carefully our paper. We will try to address your main concerns below
>
> ### About VAEBM
> Differentiating with respect to these models would indeed be relevant to this paper. We have added mentioned these works in the "related work" section.
>
> ### Overemphasizing
> Thank you for spotting this, we have tried to reformulate these sentences.
>
> ### About the variance
> We agree that other metrics might be suited to capture the geometry of the data. However, the main objective of this paper was to study the vanilla VAE framework from a fully geometric point of view. With that being said it appeared to us that the since the covariance matrices are learned from the data and favor through the posterior sampling some direction in the latent space, it is a natural choice as metric. Moreover, considering the (inverse) covariance matrices as the metric value at the embedding points was the key observation leading us to relate the Riemannian Gaussian distribution to the classic multivariate one.
>
> ### Data represented with several connected components
> We believe that since the metric relies on centroids, provided that their number is sufficient, choosing them with a k-medoids algorithm for instance would allow to deal with multi-component data. A user may also use all the training embeddings in the metric provided that the latent space dimension and the number of training points remain limited.
>
> ### VAE+GMM is almost as good as the method.
> This is true for the OASIS experiment. However, when looking at the performances obtained on the other experiments, the proposed method also appears to be more robust to data set changes as well as the number of training samples then the VAE+GMM.
>
> Please do not hesitate if you have any further questions/comments.

---

> > ### Comment · Reviewer_yrKH · 2021-11-29
> > **Regarding authors response**
> >
> > Thank you for your reply,
> >
> > Regarding VAEBM, I expected to have it as a baseline, not just to cite them. Nevermind.
> >
> > Regarding the connected component comment.
> > The same issue arises for samples "far" from the data, where no instance of the training set has been sent.
> > Could one end up outside the "data" by following the metric? If yes, what would happen? What would be the covariance there? infinite? null? in both cases, you expect to have issues with the metric.
> >
> > I appreciate the authors' effort to provide more explanation. I believe that if the appendix were included in the article, it would make a very nice journal article.
> >
> > I leave my recommendation as is.

---

### Official Review · Reviewer_NpUX · 2021-11-02

**Correctness:** 3
**Technical Novelty And Significance:** 2
**Empirical Novelty And Significance:** 2
**Recommendation:** 6
**Confidence:** 3

**Details Of Ethics Concerns:**

Using images generated from autoencoder models to train classifiers on medical images requires an understanding of the type of images generated and the possible biases that are in the dataset. I’m not an expert on this topic, but I would exercise caution when introducing dataset augmentation via VAEs (or other generative models) for training classifiers in a medical setting.

**Main Review:**

The paper provides a way of incorporating geometry into the VAE setup for generating samples that belong to this manifold. While geometry-aware sampling strategies are available in prior literature, the authors claim that these processes are time-consuming and complex. To better position this statement, I think the runtime and complexity of the proposed method need to be discussed and substantiated.

I assume the authors do not change the training process and are only proposing a better sample generation strategy - it is not clear from the current text if this is the case and if so, it is unclear what training procedure they are resorting to. The authors review Riemannian geometry concepts that are relevant to the method proposed and it is easy to follow through this part of the paper.
The toy example shows the advantage of using the proposed sample generation procedure in comparison to vanilla VAE sample generation - the proposed method generates relevant samples that maintain the simple geometric structure as in the input data.
Experiments show improvements over RAE’s, VAMP, and RHVAE in terms of FID score and classification with a VAE generated set of data. The performance of the proposed method in a low data regime looks interesting and would be interesting to have a discussion on why the proposed method is able to achieve this.

My major concern is the variability of the samples generated. I wonder if the samples generated in real dataset experiments are biased towards resampling available input data. If \rho variable used for the calculation of the metric in equation (7) is very large then the sampling process is biased towards just picking the closest input data point. I think the authors need to better support the claim on the variability of generated samples - interpolation scenarios in real images, random samples (not clear if the generated samples shown are random or cherry-picked), or even reconstructions for a given input (this would matter if the proposed method is modifying the training process as well). From the current set of visualizations and results, it is not clear if the images generated are new or just resampled input data.

Other concerns (in no particular order) are listed below.
- There is no clarity on the modification of the loss function (if any) used to train the VAE model. I assume the model is learning a mean vector and a covariance matrix for each input image - but are there any additional constraints to ensure that the matrix learned is a valid covariance matrix? More critically, what if the covariance matrix is not invertible? How are samples generated during training?

- There is no discussion on hyperparameters (\lambda, \rho) and the sensitivity of the experiments to these parameters. The supplementary material says rho is assigned by a max-min distance formulation - is this based on the entire dataset or a subset? Any reason why this is the right choice?

- In the OASIS dataset experiment, the generated dataset is made class balanced and evaluated. But the “original” classifier is trained with an unbalanced set - Is it possible to evaluate by resampling or reweighting the original training set to make them class balanced? I understand this introduces bias in the classifier but so does the experimental setup proposed.

- Any reason for dropping the RHVAE evaluation in Table 2?

- I'm not sure FID comparison with methods using prior is a fair setting - I believe it gives methods with “ex-post density estimate” an unfair advantage in reducing the gap between the posterior and the prior. This is by no means specific to this paper but I would be interested in knowing the author’s perspective on this.

The ideas and experiments by [1,2] are relevant to the proposed method and the current paper can be improved by incorporating and differentiating with respect to these recent works.

[1] Kalatzis, Dimitrios, et al. "Variational Autoencoders with Riemannian Brownian Motion Priors." International Conference on Machine Learning. PMLR, 2020.

[2] Connor, Marissa, Gregory Canal, and Christopher Rozell. "Variational Autoencoder with Learned Latent Structure." International Conference on Artificial Intelligence and Statistics. PMLR, 2021.

**Summary Of The Paper:**

The paper introduces a geometric perspective on VAEs trained with a Gaussian mixture prior (standard multivariate or learned mixture of Gaussian): the covariance matrix encoded for a given input image in a VAE corresponds to a Riemannian metric (inverse covariance) around the local neighborhood of the data point encoded.
The authors suggest sample generation in VAE  should be based on this covariance-induced distance (Mahalanobis). To obtain a single continuous metric at the end of the training, the authors suggest an interpolation based on available data and then generating data using a Monte Carlo (HMC) sampler based on this interpolated metric.
Experiments using samples generated using this method shows improvements in FID and precision-recall measures in MNIST, SVHN, CIFAR10, and CelebA datasets. The authors claim that the proposed sample generation produces good samples even with datasets of small size.


**Summary Of The Review:**

Overall, I believe the paper needs revision to better support the claims in the paper. The idea of incorporating geometry in the VAE generation procedure is interesting and has potential. I believe the authors need to improve clarity on experimental setup and evaluation, and better support the variability of samples generated. The current form of paper has a large room for improvement and I hope the authors are able to incorporate the concerns raised and better position their work.

----
Update: I appreciate the authors for answering the concerns raised and am increasing my recommendation to 6. The additional experiments and details on the algorithm in the appendix improve the clarity of the paper. The proposed sampling strategy shows improvement (FID metric) in conjunction with a simple VAE model. It might be interesting to see if the proposed sampling can be used to improve other VAE models. The experimental setup with OASIS is not as compelling (my suggestion of resampling is only a simple strategy and I believe there are better class-balancing classification methods), but highlights the shortcoming of using earlier VAE methods in this setup.

---

> ### Author Response · Authors · 2021-11-17
> **Thank your for your review!**
>
> Dear reviewer,
>
> We would like to thank you for your time and efforts reviewing our paper. Thank you also for your comprehensive review of the paper, we will try to address your concerns below:
>
> ### Runtime of the method
> We have added some discussion about the runtime of the proposed method in the paper. In addition to this, we have also added a more detailed description of the sampling process and provided the pseudo-code algorithm to better reflect its complexity in Appendix C.
>
> ### Variability of the generated samples
> We appreciate you concerns with regard to the diversity of the generated samples and the fact that they may be simply resampled train examples. However, the scores we used in the experiments are designed to assess both the fidelity/quality and diversity of the generated images when compared to the real images. In particular, the recall in the precision/recall score aims specifically at measuring the diversity of the generated samples [1]. That is true that the sampling is biased toward the centroids. However, the choice of the factor smoothing the metric ($\rho$) allows us to also prospect well the learned manifold. To better support the fact the proposed sampling procedure is not simply a resampling of the training samples, we also added in Fig.3 of the revised version the nearest image in all the reconstructions of the training images for MNIST and Celeba. It shows that the generated samples remains quite different from the reconstructions (please note that in this plot the closest reconstructed image may be different from the reconstruction of the closest training one) meaning that the scheme prospect the manifold and does not only focus on training data. We also added in Appendix D, 100 further samples generated with the proposed method on MNIST and Celeba without any cherry-picking. Finally, we also proposed in Appendix G an analysis of the diversity of the generated samples in the case where only two centroids are selected in the metric. This analysis shows that even with two centroids, the proposed method is able to generate somewhat diverse sharp samples that are neither only a resampling of the decoded centroids nor only an interpolation between the centroids since generated images present attributes that are not present in any of the reconstructed centroids images (eg. glasses). This is made possible by $\rho$ which allows prospecting around the centroids and not only *between* them.
>
> ### On the loss function
> We do not amend the training process of the vanilla VAE nor the loss. You are right. Similar to the classic framework, throughout training the VAE is learning a mean vector (here seen as a lower dimension representation of an input data) and a covariance matrix (the inverse of which is seen as the value of the Riemannian metric tensor at $\mu_i$) for each input data. As is standard in the VAE implementations, the covariance matrices are assumed to be diagonal and the encoder outputs the log of these diagonal coefficients. To ensure non singularity of the covariance matrices, the exponential is applied to the coefficients outputted by the encoder. During training, the samples are generated using the Multivariate Gaussian (which is an approximation of the Riemannian Gaussian distribution as explained in Sec. 4.2). We added some further details on the implementation of the method in Appendix.C and clarified that the training process is kept unchanged in the paper.
>
> ### Hyper-parameters
> We added some ablation studies to asses the impact of $\lambda$ and the influence of the number of centroids and their choice on the resulting sampling in Appendix G. The max-min expression for $\rho$ is computed only for the selected centroids. We admit that this choice may not be optimal but the idea behind it was to ensure some *smoothness* of the manifold while interpolating faithfully. This is discussed in Appendix.G
>
> ### OASIS experiment
> We added the RHVAE to the OASIS experiment as well as HVAE (further to Reviewer BQpw comments) and a resampled train set.
>
> ### Comparison with prior based methods
> The idea of comparing with prior based methods was to use them as baseline. As to the advantage of ex-post density estimation methods, they were indeed proposed to address the potentially poor expressiveness of the prior and should indeed be better.
>
> ### Related work
> Thank you for the references, they have been added to the "related work" section of the revised manuscript. The experiments performed in those papers are indeed interesting and would be appropriate to our work. However, we leave them for future work since we will not be able to make them fit within the 9 pages limit.
>
> Please do not hesitate if you have any further questions/comments.
>
> [1] MSM Sajjadi, O Bachem, M Lucic, O Bousquet, and S Gelly. Assessing generative models via
> precision and recall. In 32nd Conference on Neural Information Processing Systems (NeurIPS
> 2018),

---

### Official Review · Reviewer_UafD · 2021-11-05

**Correctness:** 4
**Technical Novelty And Significance:** 4
**Empirical Novelty And Significance:** 4
**Recommendation:** 8
**Confidence:** 3

**Main Review:**

#### Strengths
- The stated contributions of the paper are substantiated by the experiments.
- The proposed work is competitive and often superior to related methods on several datasets.

#### Weaknesses
- The use of non-standard English makes reading somewhat difficult.
- I would have liked to see some discussion of the cost incurred from HMC sampling
- I would have liked to see some discussion of the relationship of the proposed approach to [1-2].

[1] Rakowski, Alexander, and Christoph Lippert. "Disentanglement and Local Directions of Variance." Joint European Conference on Machine Learning and Knowledge Discovery in Databases. Springer, Cham, 2021.

[2] Rolinek, Michal, Dominik Zietlow, and Georg Martius. "Variational autoencoders pursue pca directions (by accident)." Proceedings of the IEEE/CVF Conference on Computer Vision and Pattern Recognition. 2019.

#### Possible typos:
- “...using either other regularization…” → using other regularization (p.6)
- “...4 well known databases” → “four well known datasets” (p.6)
- “visually speaking” → “qualitatively” (p.6)
- “...same models and data sets…” → same models and datasets (p.8)
- “... quite robust to the data set” → quite robust to the dataset (p.6)
- "According to us, these metrics" --> These metrics (p.9)
- "A d-dimensional manifold... d-dimensional Euclidean" --> A $d$-dimensional manifold... $d$-dimensional Euclidean (p.3)
- There are parts of figure 1 that aren’t fully described in the caption (e.g., part (a) and part (b) in top/bottom)


**Summary Of The Paper:**

The authors propose an interpretation of the VAE latent space as a Riemannian manifold and leverage this structure to design an improved sampling strategy. To construct a Riemannian manifold, the authors interpret the covariance of the posterior of each latent as a local Riemannian metric. The local metrics are interpolated to produce the latent manifold. To generate samples, the authors use HMC on the manifold specific uniform distribution constructed using local metrics.

**Summary Of The Review:**

The authors introduce a novel sampling procedure for the classical VAE that produces quantitatively and qualitatively strong results. The sampling approach leverages a novel theoretically-based interpretation of the VAE latent space structure. The paper itself is intuition building (e.g., toy example) but difficult to read in some places. It also appears the authors may have neglected a related body of work.

---

> ### Author Response · Authors · 2021-11-17
> **Thank you for your review!**
>
> Dear reviewer,
>
> We would like to thank you for your time and your comprehensive review of our paper. Thank you also for finding the contribution significant and the approach interesting and relevant to the community. Please find below the answers to your comments.
>
> ### Non standard English
> We are sorry about the English used in the paper and thank you for spotting several typos. They have been corrected in the revised manuscript and we have tried to proof read the paper to avoid new ones.
>
> ### The cost incurred from the HMC sampling
> We have added the runtime of the proposed method in the paper and compared it to both the prior based method and the mixture of Gaussians. We have also added a more detailed description of the sampling process and provided the pseudo-code algorithm to better reflect its complexity in Appendix C.
>
> ### Relationship to [1-2]
> Thanks for highlighting those papers, we have added some discussion in the related work section.
>
> Please do not hesitate if you have any further questions/comments.

---

> > ### Comment · Reviewer_UafD · 2021-11-30
> > **Thank you for your response**
> >
> > With consideration of the authors' rebuttal and comments from other reviewers, I maintain my initial rating.

---

### Author Response · Authors · 2021-11-17
**Thank you for your time and reviews!**

First of all, we would like to sincerely thank all the reviewers for their time and effort in reviewing our paper and for their constructive comments to improve it. We will respond individually to the concerns raised by the reviewers, but summarize here the main changes to the manuscript.

- We have amended Remark 1 to address the issue raised by Reviewer v2S3 regarding the choice of decoding distribution.

- Following the concern about the variability of the generated samples (raised by Reviewer NpUX) we have modified Fig.3 to add the nearest image among all the reconstructions of the training images. This aims at showing that the generated samples are not simply resampled training data. We have also added an analysis of the generated samples diversity when only two centroids are selected in the metric in Appendix.G to also further support our claims about generated samples variability. We have also discussed this in Sec. 5.1 of the revised manuscript.

- We have included in the "related work" section, all related papers mentioned by the reviewers and position our work with respect to these papers.

- We have added a small paragraph at the end of section 4.3 to better explain the approximations performed in the vanilla VAE framework and clarify the training process.

- For completeness, we have also added a model relying on normalizing flows (the Hamiltonian VAE) and aiming at improving the posterior distribution and not the prior to the experiments.

- We have added an ablation study in Appendix. G quantifying the influence of the metric hyper-parameters ($\lambda$, the number of centroids and their choice). The choice of $\rho$ is also motivated in this section. 5.1

- We have provided an overview of the runtime incurred with proposed method. in Sec.5.1 and pseudo-code algorithm reflecting the method's complexity in Appendix.C.

- Further to Reviewer v2S3 comments about reproducibility of the method, we have added a reproducibity statement at the end of the paper mentioning the relevant sections of the paper, explaining the practical details (this does not count toward the 9 page limit as per ICLR guidelines)

------------------------------------------ Edit 20 nov. 2021 ------------------------------------------

- Further to reviewer BQpw's  reply, we have also added an appendix (Appendix.I) that explains more precisely what could have been the Riemannian VAE, details what would be the main shortcomings in training this model and shows how the vanilla VAE version approximate such a model.

------------------------------------------ Edit 30 nov. 2021 ------------------------------------------

We would like to thank all the reviewers for taking the time to look at our rebuttal and their positive feedback.

Do not hesitate to contact us if have any further questions/comments.

---

### Decision · Program_Chairs · 2022-01-20

**Decision:**

Reject

**Comment:**

The paper proposes to use covariance of the approximate posterior to induce a metric on the latent space of the VAE and use it to sample from the Riemannian manifold learned by a VAE. Experiments on MNIST and CelebA show the method outperforms vanilla VAE in terms of sample quality (FID and PR scores). It is also shown to work better than baseline VAE models on a medical imaging classification task. While the reviewers have acknowledged the contributions of the paper, the novelty in the contributions and their importance/impact was seen to be rather limited. The main concern from the reviewers is -- while the paper is mainly based on the use of inverse covariance as the metric for manifold, it doesn't give a reasonable theoretical justification on it is a sensible metric that captures the intrinsic geometry of data. Authors in their response justify it as -- since the covariance matrices are learned from the data and favor through the posterior sampling some direction in the latent space, it is a natural choice as metric. This is not very convincing. A more technical justification for this will certainly make the paper more convincing. I suggest the authors to look at "Kumar, Abhishek, and Ben Poole. "On Implicit Regularization in $ β $-VAEs." International Conference on Machine Learning. PMLR, 2020" which theoretically connects inverse covariance and the Riemannian metric in Sec 5.2, and see if it can be adapted in their context.